# Ensemble Deep Learning Derived from Transfer Learning for Classification of COVID-19 Patients on Hybrid Deep-Learning-Based Lung Segmentation: A Data Augmentation and Balancing Framework

**DOI:** 10.3390/diagnostics13111954

**Published:** 2023-06-02

**Authors:** Arun Kumar Dubey, Gian Luca Chabert, Alessandro Carriero, Alessio Pasche, Pietro S. C. Danna, Sushant Agarwal, Lopamudra Mohanty, Neeraj Sharma, Sarita Yadav, Achin Jain, Ashish Kumar, Mannudeep K. Kalra, David W. Sobel, John R. Laird, Inder M. Singh, Narpinder Singh, George Tsoulfas, Mostafa M. Fouda, Azra Alizad, George D. Kitas, Narendra N. Khanna, Klaudija Viskovic, Melita Kukuljan, Mustafa Al-Maini, Ayman El-Baz, Luca Saba, Jasjit S. Suri

**Affiliations:** 1Bharati Vidyapeeth’s College of Engineering, New Delhi 110063, India; 2Department of Radiology, Azienda Ospedaliero Universitaria (A.O.U.), 09123 Cagliari, Italy; 3Department of Radiology, “Maggiore della Carità” Hospital, University of Piemonte Orientale, Via Solaroli 17, 28100 Novara, Italy; 4Advanced Knowledge Engineering Centre, GBTI, Roseville, CA 95661, USA; 5ABES Engineering College, Ghaziabad 201009, India; 6Department of Computer Science Engineering, Bennett University, Greater Noida 201310, India; 7School of Biomedical Engineering, Indian Institute of Technology (BHU), Varanasi 221005, India; 8Department of Radiology, Massachusetts General Hospital, Boston, MA 02115, USA; 9Men’s Health Centre, Miriam Hospital Providence, Providence, RI 02906, USA; 10Heart and Vascular Institute, Adventist Health St. Helena, St. Helena, CA 94574, USA; 11Stroke Monitoring and Diagnostic Division, AtheroPoint™, Roseville, CA 95661, USA; 12Department of Food Science and Technology, Graphic Era, Deemed to be University, Dehradun 248002, India; 13Department of Surgery, Aristoteleion University of Thessaloniki, 54124 Thessaloniki, Greece; 14Department of Electrical and Computer Engineering, Idaho State University, Pocatello, ID 83209, USA; 15Department of Physiology & Biomedical Engineering, Mayo Clinic College of Medicine and Science, Rochester, MN 55905, USA; 16Academic Affairs, Dudley Group NHS Foundation Trust, Dudley DY1 2HQ, UK; 17Department of Cardiology, Indraprastha APOLLO Hospitals, New Delhi 110001, India; 18Department of Radiology and Ultrasound, University Hospital for Infectious Diseases, 10000 Zagreb, Croatia; 19Department of Interventional and Diagnostic Radiology, Clinical Hospital Center Rijeka, 51000 Rijeka, Croatia; 20Allergy, Clinical Immunology & Rheumatology Institute, Toronto, ON L4Z 4C4, Canada; 21Biomedical Engineering Department, University of Louisville, Louisville, KY 40292, USA

**Keywords:** COVID, control, ResNet–UNet, transfer learning, ensemble deep learning, unseen

## Abstract

Background and motivation: Lung computed tomography (CT) techniques are high-resolution and are well adopted in the intensive care unit (ICU) for COVID-19 disease control classification. Most artificial intelligence (AI) systems do not undergo generalization and are typically overfitted. Such trained AI systems are not practical for clinical settings and therefore do not give accurate results when executed on unseen data sets. We hypothesize that ensemble deep learning (EDL) is superior to deep transfer learning (TL) in both non-augmented and augmented frameworks. Methodology: The system consists of a cascade of quality control, ResNet–UNet-based hybrid deep learning for lung segmentation, and seven models using TL-based classification followed by five types of EDL’s. To prove our hypothesis, five different kinds of data combinations (DC) were designed using a combination of two multicenter cohorts—Croatia (80 COVID) and Italy (72 COVID and 30 controls)—leading to 12,000 CT slices. As part of generalization, the system was tested on unseen data and statistically tested for reliability/stability. Results: Using the K5 (80:20) cross-validation protocol on the balanced and augmented dataset, the five DC datasets improved TL mean accuracy by 3.32%, 6.56%, 12.96%, 47.1%, and 2.78%, respectively. The five EDL systems showed improvements in accuracy of 2.12%, 5.78%, 6.72%, 32.05%, and 2.40%, thus validating our hypothesis. All statistical tests proved positive for reliability and stability. Conclusion: EDL showed superior performance to TL systems for both (a) unbalanced and unaugmented and (b) balanced and augmented datasets for both (i) seen and (ii) unseen paradigms, validating both our hypotheses.

## 1. Introduction

The COVID-19 pandemic has caused significant disruptions and health concerns worldwide and worsened traditional diseases since its emergence in late 2019. Efforts to control its spread have included non-pharmaceutical interventions, such as social distancing, mask-wearing, and quarantine measures, as well as the development and administration of vaccines. The development and administration of vaccines are effective in reducing the severity of the disease and preventing hospitalization and death [1,2,3,4,5,6].

Ongoing research and analysis are needed to better understand the effectiveness of various control measures and their impact on reducing the spread of COVID-19. There are several motivations for researching COVID-19 and its control measures. First, COVID-19 is a novel virus, and there is still much to learn about its transmission, symptoms, and long-term effects [1,7]. Second, research can help to fill these knowledge gaps and inform public health strategies. Third, COVID-19 has highlighted existing health disparities and inequities, and research can help to identify and address these issues in the context of the response to the pandemic [8,9]. Lastly, the COVID-19 pandemic has spurred artificial intelligence innovation and collaboration in fields such as medicine, epidemiology, and public health. Research can help to build on these developments and inform future responses to similar global health crises [3,6,10,11,12].

Supercomputers and graphical processing units (GPU) ease the burden of researchers in detecting medical imaging diseases [5,13,14,15], e.g., pneumonia [5,16]. Transfer learning (TL), ensemble deep learning (EDL), and hybrid deep learning (HDL) are novel methods of achieving better accuracy faster than traditional methods [17,18,19,20]. Hospitals, labs, institutes, professors, and doctors are not only adopting these new paradigms but also collaborating to help humans. There is variability in the design of studies looking at COVID-19 and its control measures [1,15,21,22,23,24], which can make it challenging to compare and draw conclusions from different studies. Some studies may have limited generalizability, as they may be conducted in specific populations and may not apply to other populations. The emergence of new variants of the virus may affect the effectiveness of existing control measures. Despite these limitations, ongoing research is critical for understanding and mitigating the impacts of COVID-19 and developing effective control measures.

Researchers are facing challenges in obtaining a COVID-19 image dataset with good volume. X-ray images are noisy, and these images could not clearly explain the infected lung areas in comparison to CT images. The CroMED and NovMED datasets have helped this research to detect infected COVID sections, but they should be processed using correct models. There are several published machine learning (ML), and deep learning (DL) models. ML models are mostly used for classification, while DLs are for feature extraction and classification. Now, it could be said that DL models are more suitable for the COVID CT image dataset than ML models [7,25]. Current DLs are already trained and tested on the ImageNet dataset with good accuracy. These models can be utilized and trained on CT images, but this would be a very slow and non-novel process. This challenge leads us to use TLs and EDLs. TLs are faster than traditional DL methodologies. EDLs are stronger than TLs. Still, researchers doubt the correct data size for DLs. The previous state of the art has proven that data augmentation and data balancing have a significant role in achieving better accuracy. Most of the AI systems are overfitted or never generalized. Such a process is called memorized rather than generalized. Such systems are not practical for clinical settings. Such systems therefore do not give accurate results when tried on unseen data sets. This is the fundamental motivation of this study. We specifically addressed a novel system design which is a cascade of three major AI systems for multicenter data set design aimed squarely at unseen analysis towards generalization. Thus, there is no system which is a combination of HDL + 5 TL + 5 EDL systems which was designed and tested on special five types of multicenter data systems of COVID + CONTROL combinations, and the design was applied to “unseen analysis” to establish generalization over memorization.

Based on the limitations in current research, we hypothesize two points to improve detection accuracy. First, the mean accuracy of EDLs is better than the mean accuracy of TLs. Second, balanced and augmented data give better results compared to data without augmentation. We studied 275 published journal and conference papers at IEEE, ScienceDirect, Springer, and MDPI. After this, we finalized EfficientNetV2M, InceptionV3, MobileNetV2, ResNet152, ResNet50, VGG16, VGG19, and Xception for our research work. These TL combinations have generated EDL models that could improve COVID-19 detection accuracy [7,25]. Five EDL models and seven TLs are consistently used over dataset combinations (DC) taken from Croatia and Italy.

The layout of this study is as follows: Section 2 presents the related literature. We discuss recent research and its accuracy on currently available datasets. Section 3 is a methodology in which the architecture and approach of research are included. The results and performance evaluation based on methodology and different performance metrics are presented in Section 4. Section 5 presents the system reliability and explainability. The critical discussion is presented in Section 6, and finally, the study concludes in Section 7.

## 2. Background Literature

The COVID-19 pandemic has led to an unprecedented global health crisis, with a significant impact on public health, and social and economic aspects of life [9,25]. One of the primary challenges that has been faced by healthcare professionals during the pandemic is the early and accurate diagnosis of COVID-19 patients. CT scans are one of the most reliable and widely used methods for the diagnosis of COVID-19 owing to their high sensitivity and specificity. With the advent of DL-based AI models, researchers have been able to develop automated diagnostic tools that can help healthcare professionals to diagnose COVID-19 patients more accurately and efficiently. Several studies have been conducted to develop and evaluate AI-based models for the diagnosis of COVID-19 using CT scans. For instance, in a study conducted by Gozes et al. [26], a DL-based model was developed and evaluated using a dataset of ~1500 CT scans. The study reported an overall sensitivity of 98% and a specificity of 92%, indicating that the model could accurately distinguish COVID-19 patients from non-COVID-19 patients. In a more recent study by Li et al. [27], a DL-based model was developed and evaluated using a dataset of 1684 CT scans obtained from 468 COVID-19 patients and 1216 non-COVID-19 patients. The model has had an overall accuracy of 91.4%, indicating that the model could accurately distinguish COVID-19 patients from non-COVID-19 patients.

Other studies have also explored the use of AI-based models for COVID-19 diagnosis using CT scans. Alshazly et al. [28] used DenseNet169 and DenseNet201 to evaluate 746 CT scan images. The authors have achieved an accuracy of 91.2%, an F1-score of 90.8%, and an AUC of 0.91 on DeseNet169 and an accuracy of 92.9%, an F1-score of 92.5%, and an AUC of 0.93 on DenseNet201. Cruz et al. [29] conducted another study using 746 CT scans. The best accuracy metrics were 82.76%, precision was 85.39%, and AUC was 0.89 using DenseNet161; the second-best model is VGG16, for which accuracy was 81.77%, precision was 79.05%, and AUC was 0.9. Shaik et al. [30], Huang et al. [31], and Xu et al. [32] also used TL-based MobileNetV2 on Dataset COVID-CT, TL-based MobileNetV2 on SARS-CoV2, and TL-based EfficientNetV2m on COVID-CT, and they achieved accuracies of 97.38%, 88.67%, and 95.66%, respectively. EDL has a major role in improving detection accuracy. There are some popular EDL paradigms on the CT dataset. Pathan et al. [33], Kundu et al. [34], and Tao et al. [35] used EDL models to achieve better accuracy in comparison to TL models. All three studies had an accuracies of more than 97%. In recent years, some authors have also used ensemble methods to detect COVID and non-COVID patients on X-ray and CT datasets [35,36,37,38,39,40]. These studies utilized at most two ensemble methods, and the datasets were also not large. Some other sections were also missing, such as with and without data augmentation results, unseen data analysis, and reason for combining all TLs. In addition to CT data classification, ensemble methods have also supported other medical sectors, i.e., breast cancer identification in early stages [41,42,43,44,45], brain tumor identification and segmentation [46,47,48,49], heart disease [49,50,51], and diabetic patient identification and treatment [52,53,54,55,56]. Statistical analysis has a vital role in finding reliable systems [57,58,59,60,61,62,63,64]. This analysis helps researchers to utilize models for real world applications [65,66,67,68,69]. Current research has utilized EDL and TL models and has accuracy above 90%. For instance, the Multi-DL RADIC model [70] demonstrated a remarkable accuracy rate of 99.4%. Similarly, the Multi-Deep system attained a high accuracy rate of 94.7% in [71], while an explainable CNN model achieved 95% accuracy [72]. Moreover, the use of transfer learning (TL) with VGG19 resulted in a 94.52% accuracy rate [73], while DL using Wavelet achieved an impressive accuracy of 99.7% [74].

The emergence of a novel era of Internet of Things (IoT) and EDL has further boosted research efforts, leading to remarkable accuracy rates. For instance, in [75], researchers achieved an accuracy rate of 98.56% using EDL in IoT. Stacked EDL also demonstrated promising results, achieving an accuracy rate of 93.57% [76]. Additionally, the best EDNC model achieved an accuracy rate of 97.75% [77], while a FUSI-CAD system based on a CNN model attained an impressive 99% accuracy rate [78]. These results highlight the effectiveness of various TL and EDL models in achieving high accuracy rates, which is crucial for COVID patients’ detection using CT dataset.

After undergoing literature review, we concluded that there is a need for a study with superior performance analysis, that is more generalized using unseen data analysis, and that is checked for reliability, and that has stability in argumentation and non-augmentation data sets. Furthermore, our COVLIAS system also underwent scientific validation.

These AI-based models need further validation, they could potentially play a crucial role in the fight against the COVID-19 pandemic, especially in resource-limited settings where access to diagnostic tools is limited. Future, we noticed that none of the models had an extended role of EDL on TL models keeping HDL-based segmentation of CT scans, which is a superior method since it undergoes quality control. Lastly, there has been no attempt to undergo generalizability or cross-domain paradigm where the testing is conducted on an “unseen dataset” taken from other clinical centers, unlike in the seen data set, where both the training and testing have been conducted from the same hospital settings. Our study exclusively addresses the “unseen analysis” and tested for the reliability and stability of the system design.

## 3. Methodology

In this section, we discuss image acquisition and data demography, overall architecture, HDL-based segmentation, TL-based classification approach, and EDL paradigm for classification. These subsections present the complete process to achieve our hypothesis.

### 3.1. Image Acquisition and Data Demographics

In this research work, we utilized two distinct cohorts from different countries. This dataset has already been validated by radiologists and doctors who are also co-authors in this paper. The first cohort, referred to as the experimental data set, consists of 80 CroMED COVID-19-positive individuals, with 57 males and the remainder female. Sample images are in Figure 1. An RT-PCR test was conducted to confirm the presence of COVID-19 in the selected cohort, with an average value of around 4 for ground-glass opacity (GGO), consolidation, and other opacities. Of the 80 CroMED patients, 83% had a cough, 60% had dyspnea, 50% had hypertension, 8% were smokers, 12% had a sore throat, 15% were diabetic, and 3.7% had COPD. Out of the total cohort, 17 patients were admitted to the intensive care unit (ICU), and 3 patients died due to COVID-19 infection [2,79,80].

The second data set included 72 NovMED COVID-19-positive individuals. Figure 2 included 47 males, and the remainder were female. An RT-PCR test was conducted to confirm the presence of COVID-19 in the selected cohort, with an average value of approximately 2.4 GGO, consolidation, and other opacities. Of the 72 NovMED patients, 61% had a cough, 9% had a sore throat, 54% had dyspnea, 42% had hypertension, 12% were diabetic, 11% had COPD, and 11% were smokers. In total, 10 patients died due to COVID-19 infection in this cohort. Figure 3 shows NovMED(control) datasets from Italy. The COVID (Croatia) dataset had dimensions of 512 × 512 and 5396 raw images, COVID (ITA) had dimensions of 768 × 768 and 5797 raw images, and control (Italy) had dimensions of 768 × 768 and 1855 raw images.

The CT dataset was acquired using a 64-detector FCT Speedia HD scanner (Fujifilm Corporation, Tokyo, Japan, 2017). The NovMED dataset, consisting of 72 COVID-19-positive individuals, was obtained from the Department of Radiology at Novara Hospital, Italy. The CT scans were performed using a 128-slice multidetector row CT scanner (Philips Ingenuity Core, by Philips Healthcare). The patients were required to have a positive RT-PCR test for COVID-19 as well as symptoms such as fever, cough, and shortness of breath. No contrast agent was administered during the acquisition, and a lung kernel of a 768 × 768 matrix together with a soft-tissue kernel was utilized to obtain a 1 mm thick slice. The CT scans were performed with a 120 kV, 226 mAs/slice detector configuration using Philips’s automated tube current modulation-Z-DOM with a spiral pitch factor of 1.08 and a 0.5 s gantry rotation time, and a 64 × 0.625 detector was considered [80]. Appendix A has more samples of the CroMED (COVID), NovMED (COVID), and NovMED (control) datasets. Data exclusion criteria for both CroMED and NOVMed dataset consisted of selection of the CT scans regions were based on the absence of metallic items and the high scan quality, free of external artefacts or blur caused from patient movement during the scanning procedure. In this group, the average patient’s CT volume had about 300 slices. During slice selection, slices with the greatest lung area were selected. Slice selection was performed by one of the senior radiologists (K.V.).

Balancing rationale: CroMED (COVID) consisted of 5396 images, while for NovMED (COVID), the data set consisted of 5797 images. NovMED (Control) consisted of 1855 images. Note that there were few control data points. The augmentation procedure consisted of increasing the COVID data two times and control data six times. Thus, the total numbers of images were changed to 10,792 (5396 × 2), 11,594 (5797 × 2), and 11,130 (1855 × 6), respectively. This was for balancing the data sets for COVID and the controls, and this made the control data sets nearly the same as the COVID data sets.

Folding rationale: The chosen sample size of COVID data was two times. This was based on the sample size computation (so-called power analysis, as discussed in the methodology section), for which the objective was to improve the accuracy. For the best accuracy, there was a need for at least 8100 images for COVID. Thus, we increased the COVID data sets by two times, i.e., to 10,792 (5396 × 2) and 11,594 (5797 × 2). Subsequently, the control was balanced by increasing the data set by six times, i.e., to 11,130 (1855 × 6). Table 1 depicts the distribution of the dataset.

### 3.2. Overall Pipeline of the System

The proposed overall architecture is portrayed in Figure 4. In this architecture, The CT machine operator and doctor have contributed to the storage of raw images of lungs for research purposes. These raw images were subjected to HDL segmentation to produce segmented data, resulting in a clear and distinct image of the lung. The latest advancements in segmentation have yielded better results when compared to raw images. We utilized both TLs and EDLs to detect COVID-19 and control cases with high accuracy, which is shown in Figure 4. In this study, we hypothesized that the mean accuracy of EDLs is superior to that of TLs. Additionally, we hypothesized that the mean accuracy of models with augmented input data, balanced with augmentation, would be greater than those without augmentation in both TLs and EDLs. We conducted scientific validation, statistical analysis, precision, recall, F1-score, and AUC to evaluate the performance of the models.

### 3.3. Hybrid Deep Learning Architecture of CT Lung Segmentation

After the data acquisition, raw input images were passed over to the HDL model for segmentation. The process of segmenting an image is breaking it up into segments, each of which corresponds to a desired class in the image. The approach that is utilized for image segmentation relies on the specific application and the characteristics of the picture that is being segmented. The study by Suri et al. [80] in the literature review has shown that HDLs are better than solo segmentation. Using the same spirit, ResNet–UNet was exclusively adopted for lung segmentation after pre-processing or quality control [81,82,83,84,85,86]. The ResNet–UNet-based HDL model is composed of 165 layers with ~16.5 million parameters. The final trained model size of the model was 188 megabytes. Using a cutoff of 80%, the model had Dice and Jaccard scores of 0.83 and 0.71, respectively.

These segmented images are the inputs for seven types of TL models and five EDL models in five different input data combinations (DC)—with and (i) without data augmentation and (ii) balanced and augmented data in predicting the presence of COVID-19 in three different datasets: CroMED (COVID), NovMED (COVID), and NovMED (control). ResNet helps in solving the vanishing gradient problem of previous models using skip connection. The convolutional neural networks (CNN) layer in ResNet brings down the sample features using stride two. UNet-based architecture helps in neutralizing the semantic segmentation problem. We have therefore used the combination of ResNet and UNet architecture to build HDL-based segmentation as shown in Figure 5. This amalgamation paradigm has effectively segmented the lungs in COVID-19 and control CT scans.

To balance the control and COVID classes, 3× augmentation of the control class was carried out using a vertical horizontal flip and 45-degree rotation. After class balancing in all five DC scenarios, data were further augmented twofold using a vertical horizontal flip and 30-degree rotation. The augmented data were analyzed over seven TLs and five EDLs in all five DC.

### 3.4. Transfer-Learning-Based Architecture for Classification

Transfer learning is one of the premier methods for classification and offers several advantages compared to DL-based classification [27,87,88]. Our seven TL models adopted were EfficientNetV2M, InceptionV3, MobileNetV2, ResNet152, ResNet50, VGG16, and VGG19, all pre-trained on the ImageNet dataset. Utilizing these TL models, we have designed false to top layers of all models and added a flatten, dense layer, dropout layer, and L2 regularizer. The flatten [89] helps to convert the multidimensional output of the previous layer to a 1-dimensional vector. It passes the values to a dense layer that has a ReLU activation function and L2 regularization with a strength of 0.001. This regularizer prevents overfitting. Dropout is another method that helps to reduce overfitting. Finally, the fully connected layer with two classes and a sigmoid activation function to the output of the previous layer. The output of the last layer represents the predicted probability for the two classes in the classification problem, COVID vs. control. All the architecture used in this work is shown in Appendix B. We have used these TL models due to their ability to bypass the long training time for scratch-based network designs [13,90].

### 3.5. Ensemble Deep Learning Architectures for Classification

The ensemble is the area of medical imaging that helps weak learners to make them stronger. We have also used the soft-max voting ensemble method. In this approach, the sum of the predicted score is used to predict the class of ensemble prediction. We have also proposed a novel Algorithm 1 to generate five EDL from TLs. EDL generators use a combination method over TL prediction score to create five EDLs from seven TLs. The accuracy of EDL is better than that of solo deep learning architecture. This is a novel algorithm for generating EDLs from TL combinations.
**Algorithm 1**: EDL generator**Result**: Combination of TL models score generates five EDLs**Input**: TL Model predicted scoreEDL = [ ];**While1** len(EDL) < 6 **do**:**While2** i in range (2,8) **do**:**While3** k in combinations, i: **do**:NewEDL = GenerateEDL(TLk,i combinations); //GenerateEDL function to generate on predicted score of TLs**If** ACC of NewEDL >ACC of contituents TLs ACC **then**EDL.append(NewEDL);**Else**Print(“Try new combination”)**End**          //End of If**End** //End of while3**End**     //End of while2**End**    //End of while1

After obtaining the segmented image, TL and EDL performed the task of accurate detection of COVID and control (Figure 6). First, these segmented data were preprocessed over all five data combinations. Parallel execution of models on original data size created a core dump (memory issue) at our GPU, which is why the input data size was reduced to 180 × 180. After that, the balancing of the COVID and control classes was performed after the augmentation of control class by 3×. Once the data were balanced, we augmented the data by 2× to increase the size of the data. Second augmentation was also performed to check the augmentation effect. Seven TLs were used for feature extraction, and the sigmoid function was used for binary classification. TL combinations generate EDL and EDL uses softMAX voting on the predicted score for detection of COVID and control. We have also performed balancing and augmentation on data.

### 3.6. Loss Function

Cross-entropy (CE)-loss functions are frequently used for two or more than two DL models. CE-loss, $\propto_{CE}$, is dependent on the probability of the AI model $p_{i}$ and the gold standard label 1 and 0 by $g_{i}$ and $\left(1-g_{i}\right)$, respectively, as shown in Equation (1).
(1)$$\propto_{CE}=-\left[\left(g_{i}\times \log p_{i}\right)+\left(1-g_{i}\right)\times \log\left(1-p_{i}\right)\right]$$

### 3.7. Performance Metric

We have used true positive (TP), true negative (TN), false positive (FP), and false negative (FN) to estimate the various performance evaluation metrics. These are accuracy (ɳ) (Equation (2)), recall (Ɍ) (Equation (3)), precision (ɳ) (Equation (4)), and F1-score (Ƒ) (Equation (5)). After calculating the accuracy of TL and EDL models, we calculated the mean accuracy of TL (${\bar{ɳ}}_{\mathrm{TL}}$) in Equation (6) and the mean accuracy of EDL (${\bar{ɳ}}_{\mathrm{EDL}}$) in Equations (7) and (8). In these equations, “n” is the number of TLs, and “N” is the number of EDLs. Dice and Jaccard are also calculated based on Equations (9) and (10), where ƶ is a set of wanted items and ƴ set of found items. The probability curve ROC (receiver operating characteristics) and degree of separability AUC (area under the curve) have also been calculated for each model. In the standard deviation (σ), each value from the population is denoted by x¡ and µ, population mean. N is the size of the population.
(2)$$ɳ=\frac{TP+TN}{TP+FP+FN+TN}$$
(3)$$Ɍ=\frac{TP}{TP+FN}$$
(4)$$Ƥ=\frac{TP}{TP+FN}$$
(5)$$Ƒ=2\;\times \;\left(\frac{Ƥ\times Ɍ}{Ƥ+Ɍ}\right)$$
(6)$${\bar{ɳ}}_{TL}=\frac{\sum_{i=1}^{K}{ɳ}_{TL\;i}}{K}$$
(7)$${\bar{ɳ}}_{EDL}=\frac{\sum_{i=1}^{M}{ɳ}_{EDL\;i}}{M}$$
(8)$$\sigma =\sqrt{\frac{\sum_{i=1}^{N}{\left(x_{i}-\mu \right)}^{2}\;}{N}}\;$$
(9)$$ẞ\left(ƴ,ƶ\right)=2\frac{\left|ƴ\right|\left|ƶ\right|}{\left|ƴ\right|+\left|ƶ\right|}$$
(10)$$ℐ\left(ƴ,ƶ\right)=\frac{ẞ\left(ƴ,ƶ\right)}{2-ẞ\left(ƴ,ƶ\right)}$$

### 3.8. Experimental Protocol

#### 3.8.1. Five Data Combinations

For the robust design of the classification system, we designed five types of data combination scenarios. This is based on training and testing data using taken from two countries—namely, Croatia and Italy.

DC1: Training validation and testing using both CroMED (COVID) and NovMED (control).DC2: Training validation and testing on both NovMED (COVID) and NovMED (control).DC3: Training validation using CroMED (COVID) and NovMED (control) and testing on NovMED (COVID) and NovMED (control).DC4: Training validation using NovMED (COVID) and NovMED (control) and testing on CroMED (COVID) and NovMED (control).DC5: Training validation and testing on mixed data in which COVID CT scans from Croatia and Italy are mixed; the control of Italy was used.

#### 3.8.2. Experiment 1: Transfer Learning Models using Lung Segmented Data

This experiment consists of running seven types of TL models—namely, EfficientNetV2M, InceptionV3, MobileNetV2, ResNet152, ResNet50, VGG16, and VGG19—all pre-trained on the ImageNet dataset for classification of segmented lung data into COVID vs. controls. The lung segmentation was conducted using ResNet–UNet, and segmented images were input for TLs. The experiment highlights the effectiveness of using TL for improving the accuracy of models on HDL segmented data. The lung segmented data were split with a ratio of 80, 10, and 10 for training, validation, and testing, respectively. Models were saved after training and validation and later tested over 10% of the dataset under five input data combinations. These TLs further predict COVID and control.

#### 3.8.3. Experiment 2: Ensemble Deep Learning for Classification

Here, TLs are combined to design EDL to achieve even higher accuracy in detecting COVID-19 vs. control [91,92,93,94,95,96,97,98]. In DC1 five EDLs have been generated on TL models. EDL1: VGG19 + VGG16, EDL2: InceptionV3 + VGG19, EDL3: VGG19 + EfficientNetV2M, EDL4: InceptionV3 + EfficientNetV2M, EDL5: ResNet50 + EfficientNetV2M. Using DC2, EDL1: ResNet50 + ResNet152, EDL2: VGG16 + EfficientNetV2M, EDL3: VGG19 + EfficientNetV2M, EDL4: VGG16 + MobileNetV2, EDL5: VGG19 + MobileNetV2. Using DC3 EDL1: ResNet50 + MobileNetV2, EDL2: ResNet50 + InceptionV3, EDL3: InceptionV3 + VGG19, EDL4: InceptionV3 + MobileNetV2 + ResNet152, EDL5: InceptionV3 + EfficientNetV2M + ResNet152. EDL1: EfficientNetV2M + ResNet50, EDL2: MobileNetV2 + ResNet50, EDL3: ResNet50 + ResNet 152, EDL4: ResNet152 + EfficientNetV2M, MobileNetV2 + VGG19 for DC5.

#### 3.8.4. Experiment 3: Effect of EDL Classification over TL Classification with Augmentation

This experiment is to show the effect of EDL classification over TL classification on unaugmented data and augmented data [99,100,101,102,103]. Mean EDL accuracy and mean TL accuracy verified after the balance and augmentation.

#### 3.8.5. Experiment 4: Unseen Data Analysis

Training on one combination of data and testing on another combination of data were experimented with here. We analyzed the models’ performance on unseen data to evaluate their generalizability [104,105,106,107,108,109,110,111]. The results showed that the models performed well on unseen data, indicating their potential for real-world applications. Input data scenarios DC3 and DC4 are examples of unseen data analysis.

### 3.9. Experimental Setup

We used Idaho State University (ISU) GPU cluster for executing all models using DC1 to DC5. Tensorflow 2.0 libraries helped us to design the software, and results were also evaluated using MedCalc v12.5 statistical software [112,113,114,115,116,117]. Common hyperparameters in TL models are Optimizer: Adam, Learning rate: 0.0001, Loss: function categorical_cross-entropy, Regularizer: L2 (0.01), Dropout: 0.5, Batch Size: 32, Classification activation function: Sigmoid, Other layer activation function: Relu, and epoch: 25.

### 3.10. Power Analysis

We calculated the sample size using the conventional method [118,119,120]. The formula for calculating the sample size is represented by n, is as follows: (11)$$n=\left[{\left(z^{*}\right)}^{2}\times \left(\frac{\tilde{p}\left(1-\tilde{p}\right)}{{\mathrm{MoE}}^{2}}\right)\right]$$
where z* is the z-score corresponding to the desired level of confidence, MoE is the margin of error (half the width of the confidence interval), and is the estimated proportion of the characteristic in the population. Using MedCalc software, we calculated the required values and substituted them into Equation (11). We need a sample size of at least 8100 to estimate the proportion of the characteristic in the population with a 95% confidence interval of 0.963 to 0.978 and an MoE of 0.0075.

## 4. Results and Performance Evaluation

To verify both hypotheses, we conducted four experiments on five DC scenarios. ResNet–UNet, a hybrid deep learning model, was used to segment the raw data. CroMED (COVID), NovMED (COVID), and NovMED (control) raw images are there along with the segmented images. We randomly selected four sample images from CroMED (COVID) and passed them through ResNet–UNet. The output segmented images have been placed below the raw images in Figure 7. With the same approach, NovMED (COVID) and NovMED (control) information is stored in the same diagram. All five DC have utilized the seven transfer learning models and five ensemble deep learning models over CroMED (COVID), NovMED (COVID), and NovMED (control). The seven TLs are EfficientNetV2M, InceptionV3, MobileNetV2, ResNet152, ResNet50, VGG16, and VGG19, and a combination of TL models with soft-voting ensemble methods generates EDL models. Training accuracy and loss plots for the ResNet–UNet on each epoch is shown in Figure 8.

### 4.1. PE for HDL Lung Segmentation

Figure 9 depicts a cumulative frequency plot for Dice (left) and Jaccard (right) for ResNet–UNet when computed against the medical doctor (MD) 1. The correlation coefficients and BA plots for MD1 and MD2 are shown in Figure 10 and Figure 11. The correlation coefficient graph depicts the relationship strength between ResNet–UNet and doctors’ views. The BA plot shows the compatibility between ResNet and UNet. After the segmentation of images, TL and EDL models utilize this segmented image for classification. We have decided on five scenarios for classification to prove our hypothesis. The evaluation metrics used to compare the models include mean accuracy (Mean ACC), standard deviation (SD), mean predicted score (Mean PR), area under the curve (AUC), *p*-value, precision, recall, and F1 score.

### 4.2. Results of Experiment 1: Transfer Learning Models using Lung Segmented Data

In Experiment 1, we performed the TLs operations using ResNet–UNet segmented data. Following are the detailed results for all five DC scenarios.

DC1 results: Table 2 and Figure 12 show that the best accuracy of 97.93% without augmentation and 99.93% with augmentation is shown by MobileNetV2. The mean accuracy of all seven TLs without augmentation is 93.91% and is 97.03% with augmentation. For TL6 (VGG16), the accuracy improves from 90.20% (before augmentation) to 95.61% (after augmentation), so the improvement was 5.41% using DC1 data combination. TL2 (Inception V3) had an accuracies of 93.60% (before augmentation) and 93.97% (after augmentation), so the improvement was 0.37%. Therefore, we see that augmentation has different effects on TL-based classifiers. It is more pronounced in TL6, unlike in TL2. Table 3 shows the COVID precision are significantly increased or comparable after balancing data.DC2 results: Table 4 and Figure 13 show that the best accuracy of 90.84% is achieved by InceptionV3 without augmentation, and the best with augmentation of 93.92% is achieved by EfficientNetV2M. The mean accuracy of all seven TLs without augmentation is 84.41% and is 89.85% with augmentation. TL4 (ResNet152), the accuracy improves from 78.16% (before augmentation) to 87.40% (after augmentation) when using DC2 data combination, so the improvement was 11.82%. TL6 (VGG16) had accuracies of 85.6% (before augmentation) and 84.05% (after augmentation), so there was no improvement. Therefore, we see that augmentation has different effects on TL-based classifiers. It is more pronounced in TL4, unlike in TL6. Table 5 shows the effect of augmentation in COVID precision, recall and F1-score. These are significantly increased or comparable after balancing data.DC3 results: Table 6 and Figure 14 show that the best accuracies of 85.40% without augmentation and 91.41% with augmentation are achieved by EfficientNetV2M. The mean accuracy of all seven TLs without augmentation is 72.90% and is 82.355% with augmentation. For TL5 (ResNet50), the accuracy improves from 67.17% (before augmentation) to 80% (after augmentation) when using DC3 data combination, so the improvement was 19.10%. TL2 (InceptionV3) had accuracies of 67.58% (before augmentation) and 76.43% (after augmentation), so the improvement was 13.09%. Therefore, we see that augmentation has different effects on TL-based classifiers. It is more pronounced in TL5, unlike in TL2. Augmentation and balancing effects are visible in Table 7. It shows that better results can be achieved after balancing the data.DC4 results: Table 8 and Figure 15 show that the best accuracies of 69.40% without augmentation and 81.05% with augmentation are shown by VGG19. The mean accuracy of all seven TLs without augmentation is 47.85% and is 70.39% with augmentation. The augmentation effect was also visible with TL3 (MobileNetV2), which had a lowest accuracy of 27.97% before augmentation and 52.68% after the augmentation, so the improvement is 92.5%. Table 9 has been presented to show augmentation effect for precision, recall and F1-score.DC5 results: Table 10 and Figure 16 show that the best accuracy of 95.10% is achieved by InceptionV3 without augmentation, and 95.28% is achieved by VGG16 with augmentation. The mean accuracy of all seven TLs without augmentation is 91.22% and is 93.76% with augmentation. TL6 (VGG16), the accuracy improves from 86.81% (before augmentation) to 95.28% (after augmentation) when using DC5 data combination, so the improvement is 9.75%. TL3 (MobileNetV2) has an accuracies of 92.95% (before augmentation) and 89.07% (after augmentation), so there is no improvement. Therefore, we see that augmentation has different effects on TL-based classifiers. It is more pronounced in TL6, unlike in TL3. In the most of TL models, improvement of precision, recall and F1-score can be seen Table 11, after the balancing and augmenting the data.Table 3, Table 5, Table 7, Table 9 and Table 11 also show the precision and a comparison to Experiment 1, which presents the verification of Hypothesis 2 that data augmentation helps in improvement in the performance of TL model. *p*-value based on the Mann–Whitney test was used for all data combinations.

### 4.3. Results of Experiment 2: Ensemble Deep Learning for Classification

In Experiment 2, we performed the EDL operations for accurate classification of COVID and control. These EDLs are created using TL models. Following are the detailed results for all five DC scenarios.

DC1 results: Table 12 and Figure 17 show that the mean accuracy of all EDLs without augmentation is 95.05% and is 97.07% with augmentation.DC2 results: Table 13 and Figure 18 show that the mean accuracy of all EDLs without augmentation is 87.63% and is 92.70% with augmentation.DC3 results: Table 14 and Figure 19 show that the mean accuracy of all EDLs without augmentation is 75.88% and is 80.98% with augmentation.DC4 results: Table 15 and Figure 20 show that the mean accuracy of all EDLs without augmentation is 59.99% and is 79.22% with augmentation.DC5 results: Table 16 and Figure 21 show that the mean accuracy of all EDLs without augmentation is 93.39% and is 95.64% with augmentation.

### 4.4. Results of Experiment 3: EDL vs. TL Classification with Augmentation

In Experiment 3, we verified the effect of augmentation in EDLs over TLs in all five DC scenarios. Figure 22 shows results in unaugmented data, and we observed an accuracy improvement in EDLs over TLs of 5.54%. Similarly, Figure 23 shows an accuracy improvement of 2.82% in EDLs over TLs with balanced and augmented data. This verifies Hypothesis 1.

### 4.5. Results of Experiment 4: Unseen Data Analysis

In Experiment 4, we performed unseen data analysis. In the DC3 scenario, training was performed on CroMED (COVID) and testing on NovMED (COVID). Similarly, in DC4 scenarios, training was performed on NovMED (COVID) and testing on CroMED (COVID). As shown in Figure 22 and Figure 23, we observed that even in unseen data analysis, both of our hypotheses are proven to be correct.

The comparative graph of mean TL accuracy and mean EDL accuracy proves both of our hypotheses. First, the mean accuracy of EDLs is better than the mean accuracy of TLs. Second, balanced and augmented data give better results compared to those without augmentation. We have also presented the standard deviation, mean predicted score, AUC, and *p*-value for all input data scenarios. DC1, DC2, DC3, DC4, and DC5 TL models with data augmentation and balance improved mean accuracy by 3.32%, 6.56%, 12.96%, 47.1%, and 2.78%, respectively. Similarly, the five EDLs’ accuracies increased by 2.12%, 5.78%, 6.72%, 32.05%, and 2.40%, respectively.

### 4.6. Receiver Operating Charaterstics

We calculated the AUC from ROC graphs for our model to check explainability. Figure 24, Figure 25, Figure 26, Figure 27 and Figure 28 show TL1: EfficientV2M, TL2: InceptionV3, TL3: MobileNetV2, TL4: ResNet152, TL5: ResNet50, TL6: VGG16, and TL7: VGG19.ROC for input data scenarios DC1, DC2, DC3, DC4, and DC5, respectively.

Overall, the results show that deep learning models based on transfer learning and ensemble methods achieve high accuracy in detecting COVID-19. Among the transfer learning models, MobileNetV2 outperforms the other models in terms of accuracy and AUC in all five cases. In addition, the ensemble models show better performance than individual transfer learning. Similar to TL ROC, EDL’s ROC can also be generated. All EDLs AUC-ROC for all five data combinations is already discussed in the result Section 4.2 tables. One of the data combinations, DC1, with augmentation ROC is depicted in Figure 29. It shows that at most of the AUC points of EDLs are better than or equal to their constituents. Data combinations of two to five scenarios ROC are in Appendix C.

## 5. System Reliability

### Statistical Test

Paired *t*-test, Mann–Whitney, and Wilcoxon tests were performed to check the reliability of the system in all five IDS. The *p*-value was less than 0.0001 in all five input data scenarios cases. This shows that our proposed system is highly reliable for real-world applications. The test has been performed using Python and MedCalc software. Result section tables also have stored *p*-values on the Mann–Whitney test for all TLs and EDLs using DC1, DC2, DC3, DC4, and DC5. Similarly paired t-test and Wilcoxon tests were also performed, and their summaries have been stored in Table 17 and Table 18. Table 17 shows the three statistical tests (paired *t*-test, Mann–Whitney, and Wilcoxon tests) for seven TL models (EfficientV2M, InceptionV3, MobileNetV2, ResNet152, ResNet50, VGG16, and VGG19). As seen in Table 17, all the TL models (TL1-TL7) exhibit *p*-values <0.0001. This clearly demonstrates the TL models’ reliability and stability as per the definition null hypothesis. Table 18 presents the three statistical tests (paired *t*-test, Mann–Whitney, and Wilcoxon tests) for five EDL models (EDL1-EDL5). As seen in Table 18, all the EDL models exhibit *p*-values <0.0001. This clearly demonstrates the EDL model’s reliability and stability as per the definition null hypothesis. Note that our results are consistent with our previous studies [80,121,122,123,124,125,126].

## 6. Discussion

The proposed system has been trained on multicenter data using different acquisition machines and incorporated superior quality control techniques, class balancing using augmentation, and ResNet–UNet HDL segmentation. It uses seven types of TL classifiers and five types of EDL-based fusion to make accurate predictions. The model employed uniquely designed data systems, a generalized cross-validation protocol, and performance evaluation of HDL segmentation, TL classification, and EDL systems. It was also tested for reliability analysis and stability analysis and benchmarked against previous TL and EDL research work.

### 6.1. Principal Findings

Explainable transfer learning (TL) and ensemble deep learning (EDL) models accurately predicted the presence of COVID-19 in Croatian and Italian datasets and justified both hypotheses. This architecture presented the behavior of models on data augmentation and balancing. TL accuracy with augmentation and balancing overperformed compared to that without augmentation. Some TL and EDL models outperformed the benchmark in most cases in accuracy, precision, recall, F1 score, and AUC. The proposed method, which uses ResNet–UNet for segmentation and TL and EDL models for classification, is a promising approach for identifying COVID-19 in CroMED (COVID), NovMED (COVID), and NovMED (control). It is a novel approach that uses HDL segmentation for ensemble-based classification. Overall, these findings suggest that ensemble deep learning models can be useful tools for identifying COVID-19 and controlling its spread. Unseen analysis in data combinations two and three show that this infrastructure could be used for real world application. These TL and EDL results have proven that, and the novelties can be summarized as (i) implementation of ResNet–UNet-based HDL segmentation; (ii) executing seven types of TL classifiers, design of five types of EDL-based fusion; (iii) design of five types of data systems; (iv) generalized COVLIAS system design using unseen data; (v) tested for reliability analysis; (vi) tested for stability analysis. The methods applied in this study have created an effective and robust system that has better performance metrics in comparison to existing published models.

### 6.2. Benchmarking

We studied several papers and sorted some recent papers for benchmarking. These papers include the COVID-CT dataset and the SARS-CoV-2 dataset [127,128,129,130,131,132,133,134,135,136,137]. Our proposed models have used the CroMED (COVID), NovMED (COVID), and NovMED (control) datasets. Seven state-of-the-art transfer learning models, including DenseNet201, DenseNet169, DenseNet161, DenseNet121, VGG16, MobileNetV2, and EfficientNetV2M, have been used on the COVID dataset and compared with our best proposed model on MobileNetV2. We evaluated the models based on their accuracy, precision, recall, F1 score, *p*-value, and AUC and compared the results to the previous benchmark studies. Our experimental results showed that our proposed method, which used MobileNetV2 on Dataset 1 (CroMED (COVID) and NovMED (control)), outperformed all other models, with an accuracy of 99.99%, precision and recall of 100%, F1 score of 100%, and AUC of 1.0. The second-best model was DenseNet121 by Xu et al. [32], which achieved an accuracy of 99.44% on the COVIDx-CT 2A dataset. It is presented in Table 19. We have also compared our best TL with other existing models proposed by Alshazly et al. [28], Cruz et al. [45], Shaik et al. [30], and Huang et al. [31], who achieved accuracies of 92.9%, 82.76%, 97.38%, and 95.66%, respectively. Our results demonstrate the effectiveness of TL in developing accurate and efficient models for COVID-19 diagnosis using CT images. Our findings highlight the importance of using larger and more diverse datasets for training DL models for medical image analysis. Like the TL model comparison, we have also compared our proposed EDL models to state-of-the-art EDL models. We evaluated the models based on their accuracy, precision, recall, F1-score, and AUC and compared the results to the previous benchmark studies. The ensemble model, a combination of ResNet152 + MobileNetV2, outperformed all other models, with an accuracy of 99.99%, precision and recall of 100%, F1 score of 100%, AUC of 1.0, and p-value of less than 0.0001. The second-best model, with an accuracy of 99.05% and an F1 score of 98.59%, was proposed by Toa et al. [35]. Other ensemble models are also shown in Table 20 and are quite lower than our proposed model. Other EDL models were proposed by Pathan et al. [33], Kundu et al. [34], Cruz et al. [29], Shaik et al. [30], Khanibadi et al. [138], Lu et al. [139], and Huang et al. [31]. We also performed scientific validation that is missing in other models.

These TL and EDL results have proven that the novelties—ResNet–UNet HDL segmentation+ seven types of TL classifier + design of five types of EDL-based fusion + design of five types of data systems + generalized COVLIAS system design using unseen data + tested for reliability analysis + tested for stability analysis—applied in this study have created an effective and robust system that has better performance metrics compared to existing published models.

### 6.3. A Special Note on EDL

Ensemble-based models can be effective in addressing some of the limitations and weaknesses in current published research work on COVID-19 and its control measures. Ensemble-deep-learning-based models are deep learning models that combine multiple models to make more accurate predictions than any single model alone. This approach can improve the generalizability and robustness of predictions, which can be particularly useful in the context of COVID-19 research. Ensemble models always survive when the amalgamation of features or predicted score improves accuracy. If there is a bias in data, then EDL survival is difficult.

### 6.4. Strengths, Weaknesses, and Extensions

The study compares seven transfer learning and five ensemble deep learning models in predicting the presence of COVID-19, providing a comprehensive evaluation of different approaches. This work uses data augmentation and balanced data to improve the performance of the models, which can be a valuable technique in improving model accuracy. Our research outperforms the benchmark results in most cases, indicating that the proposed models are effective in predicting the presence of COVID-19. The study only uses three datasets, CroMED (COVID), NovMED (COVID), and NovMED (Control), which limits the generalizability of the results. It does not compare the proposed models to other COVID-19 prediction models that may have been developed outside of the benchmark studies. The work could investigate the impact of other segmentation methods on the accuracy of the models. Transformers can also be added for segmentation and detection of COVID-19 [140,141,142,143,144]. While the system is generalized, the system lacks explainability of the AI models, so-called explainable AI (XAI) models. The system lacks the role of superposition of heatmaps on the lung CT images, which can tell where COVID-19 lesions are present, especially using these TL models applied to the HDL segmented lung outputs. Previous systems have used heatmaps [5,121,122] but not in the cascaded framework of HDL + TL + EDL in the multicenter paradigm. Since the field of immunology brings discussions on lung damage causing different kinds of pneumonia, the current paradigm of COVID/control binary classification can be extended to multiclass framework. Our group has several studies which followed multiclass classification using AI framework [145,146,147,148,149]. Our system can therefore be extended as we acquire clinical data for different kinds of pneumonia.

## 7. Conclusions

In this research work, we had two hypotheses. First, that mean TL accuracy with augmentation is better than without augmented data, which was proven in all five input data scenarios. DC1, DC2, DC3, DC4, and DC5 TL models with data augmentation and balance improved mean accuracy by 3.32%, 6.56%, 12.96%, 47.1%, and 2.78%, respectively. Second, that weaker learners would be stronger in the ensemble process, and that mean EDL accuracy would over mean TL, which is visible in performance evaluation. Explainable transfer learnings have generated ROCs. These are useful for identifying better models. Three statistical tests have shown *p*-values of less than 0.0001 for all models. This indicates that the system is highly reliable. We have also compared our results to the benchmark results on the COVID dataset. The ensemble model, a combination of ResNet152 and MobileNetV2, outperformed all other models, with an accuracy of 99.99%, precision and recall of 100%, F1 score of 100%, AUC of 1.0, and *p*-value of less than 0.0001. The second-best benchmark model has 99.05% accuracy and a 98.59% F1 score. Our findings have not only supported both hypotheses, but the proposed methodology also outperforms benchmark performance indicators.

Some future works can also be implemented. We have performed a soft-max voting method in the ensemble process; fusion of features before the prediction is also an option. Statistical tests will confirm system reliability, and a heatmap of the ensemble model could also be generated.

## Figures and Tables

**Figure 1 diagnostics-13-01954-f001:**
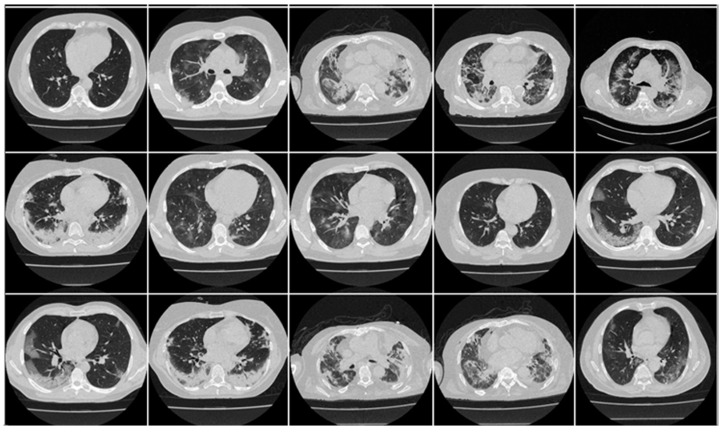
Raw “COVID-19 CT slices” patient images taken from CroMED dataset.

**Figure 2 diagnostics-13-01954-f002:**
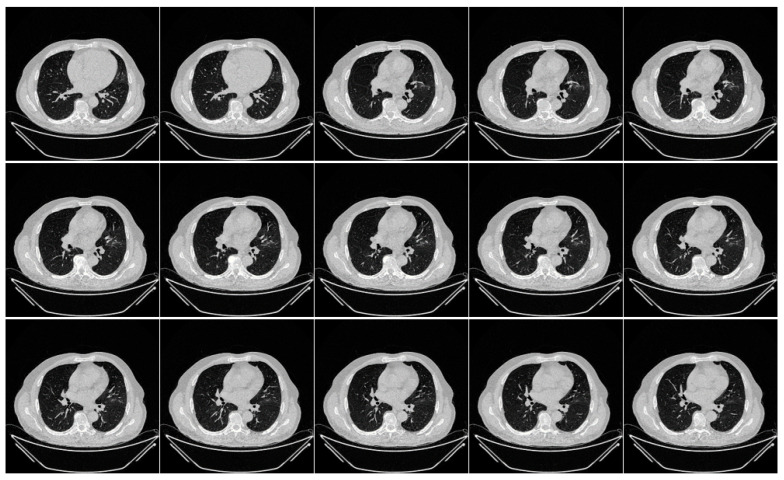
Raw “COVID-19 CT slices” images taken from NovMED dataset.

**Figure 3 diagnostics-13-01954-f003:**
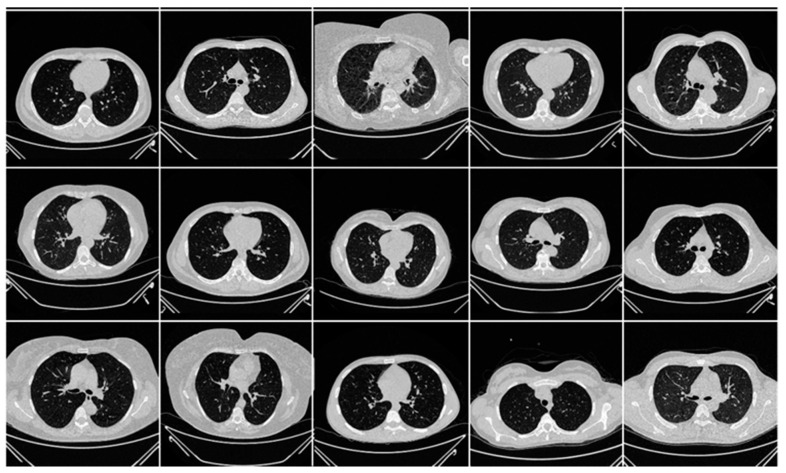
Raw “Control CT slices” images taken from NovMED dataset.

**Figure 4 diagnostics-13-01954-f004:**
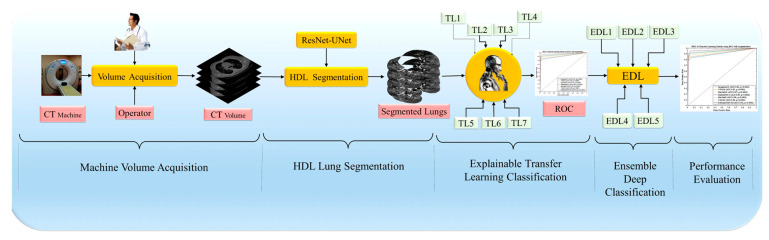
Overall pipeline consisting of CT volume acquisition, HDL-based segmentation, transfer learning (TL1–TL7), and ensemble-deep-learning-based (EDL1–EDL5) classification.

**Figure 5 diagnostics-13-01954-f005:**
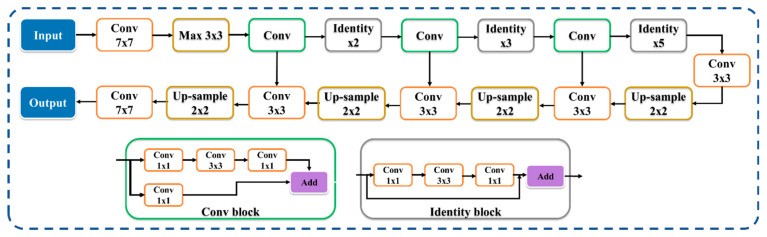
ResNet–UNet HDL architecture for lung segmentation [80].

**Figure 6 diagnostics-13-01954-f006:**
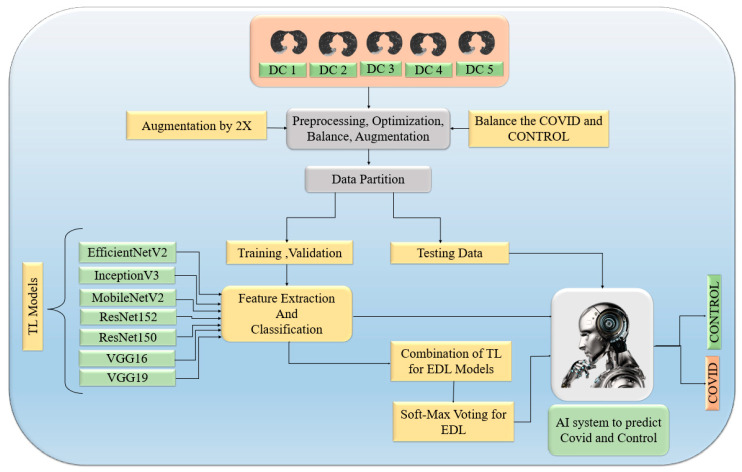
TL and EDL classification process on segmented images.

**Figure 7 diagnostics-13-01954-f007:**
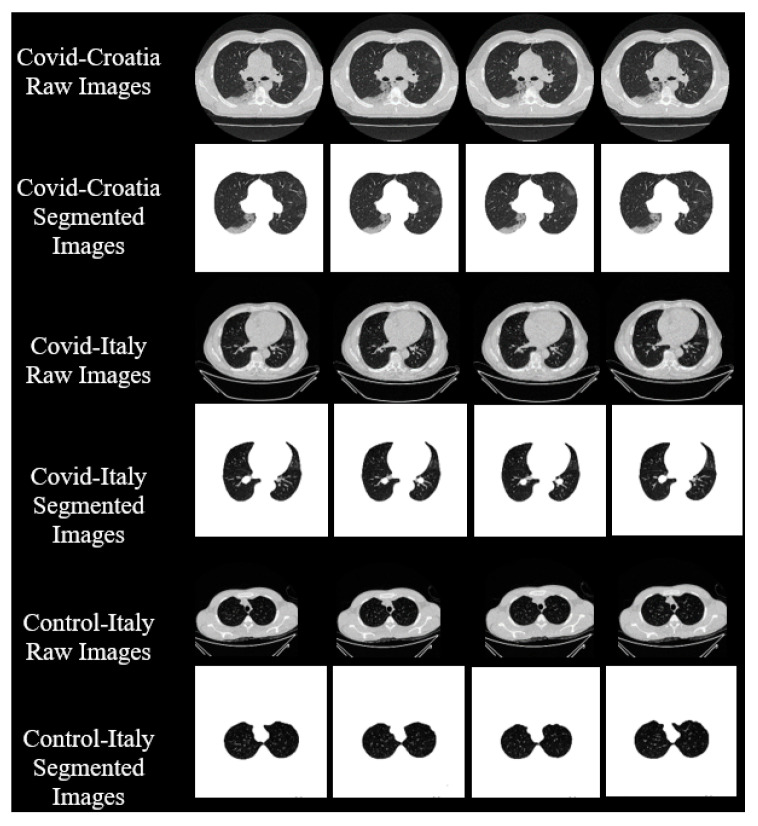
Raw images and corresponding segmented images after ResNet–UNet.

**Figure 8 diagnostics-13-01954-f008:**
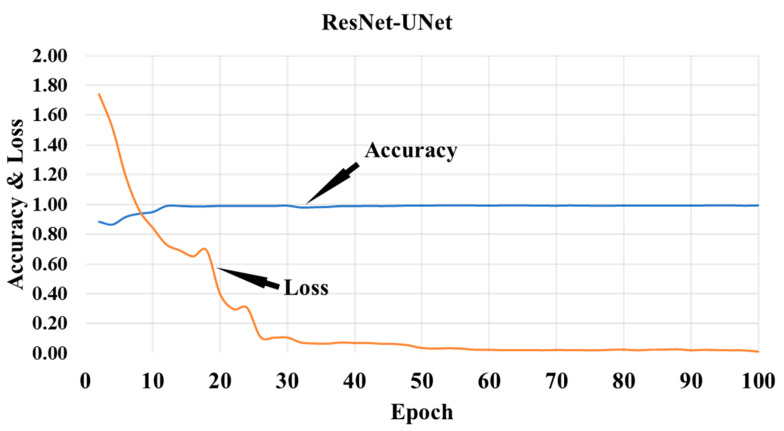
Training accuracy and loss plot for ResNet–UNet.

**Figure 9 diagnostics-13-01954-f009:**
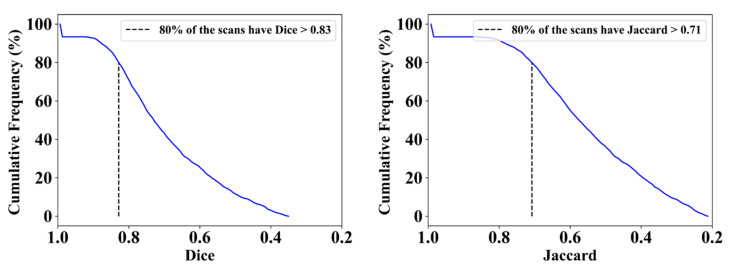
Cumulative frequency plot for Dice (**left**) and Jaccard (**right**) for ResNet–UNet when computed against MD 1.

**Figure 10 diagnostics-13-01954-f010:**
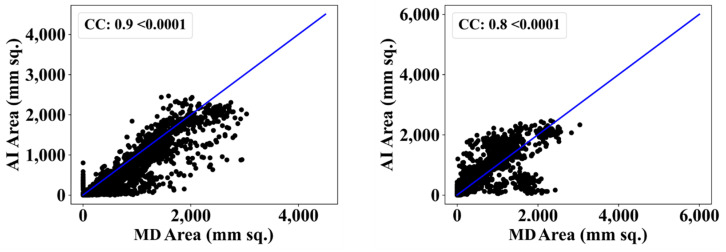
Correlation coefficient plot for left: ResNet–UNet vs. MD 1 and right: ResNet–UNet vs. MD 2.

**Figure 11 diagnostics-13-01954-f011:**
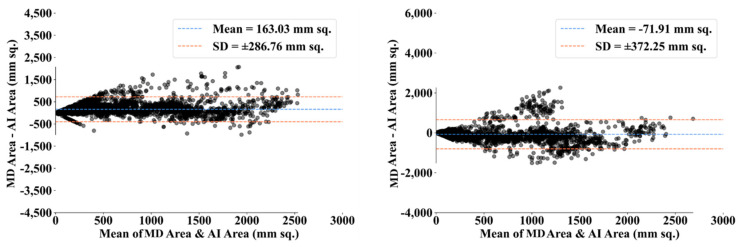
BA plot for ResNet-UNet using MD 1 (**left**) vs. MD 2 (**right**).

**Figure 12 diagnostics-13-01954-f012:**
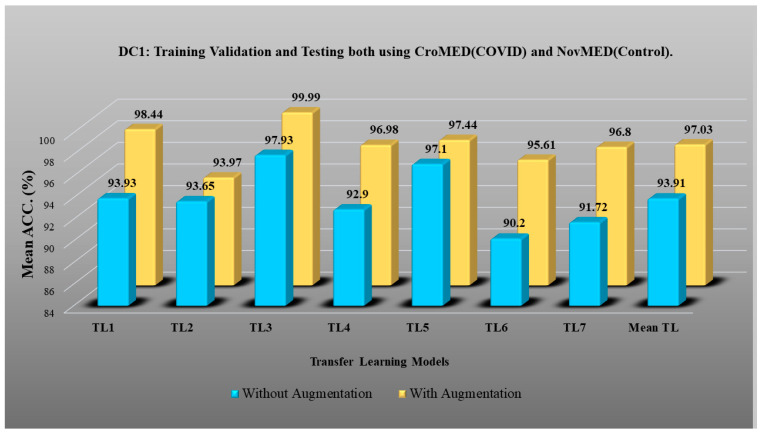
Comparison of mean TL accuracy with/without augmentation. TL1: EfficientV2M, TL2: InceptionV3, TL3: MobileNetV2, TL4: ResNet152, TL5: ResNet50, TL6: VGG16, TL7: VGG19 using DC1.

**Figure 13 diagnostics-13-01954-f013:**
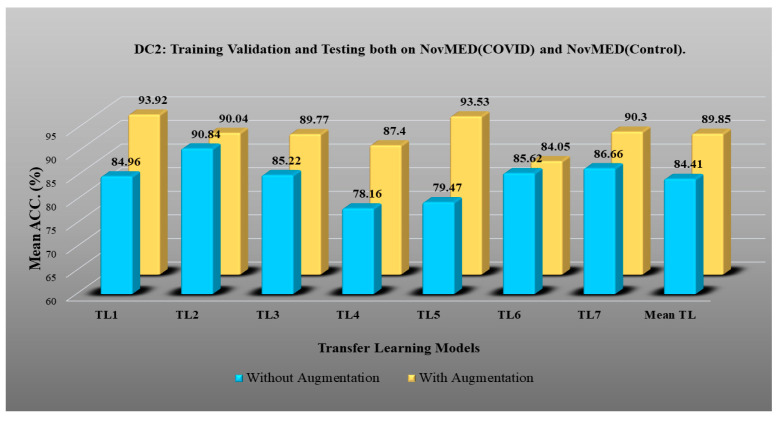
Comparison of mean TL accuracy with/without augmentation. TL1: EfficientV2M, TL2: InceptionV3, TL3: MobileNetV2, TL4: ResNet152, TL5: ResNet50, TL6: VGG16, TL7: VGG19 using DC2.

**Figure 14 diagnostics-13-01954-f014:**
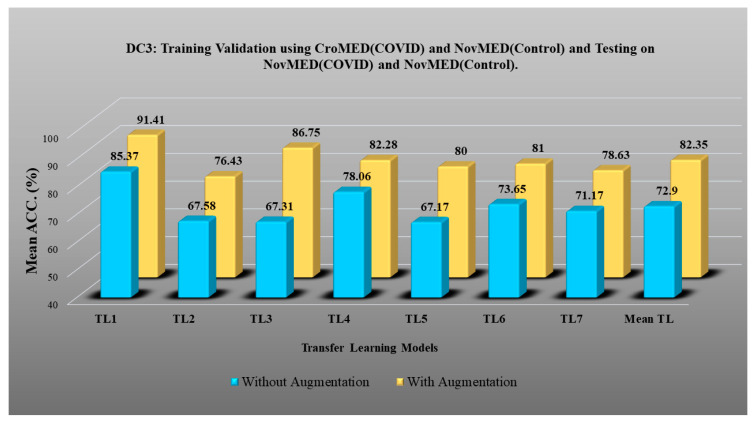
Comparison of mean TL Accuracy with/without augmentation. TL1: EfficientV2M, TL2: InceptionV3, TL3: MobileNetV2, TL4: ResNet152, TL5: ResNet50, TL6: VGG16, TL7: VGG19 using DC3.

**Figure 15 diagnostics-13-01954-f015:**
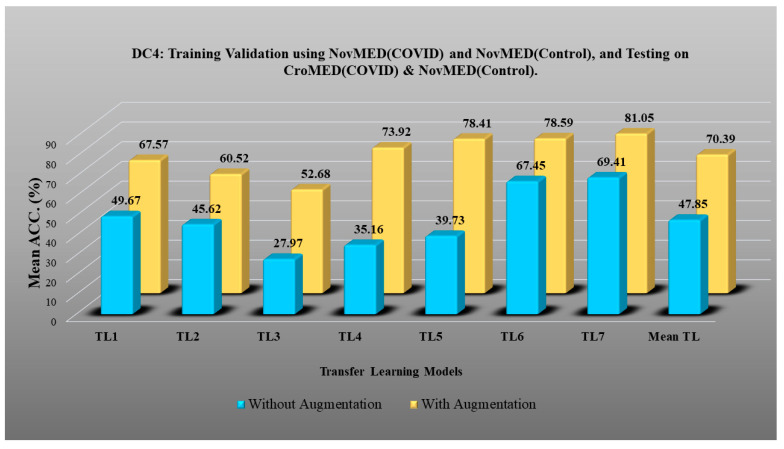
Comparison of mean TL accuracy with/without augmentation. TL1: EfficientV2M, TL2: InceptionV3, TL3: MobileNetV2, TL4: ResNet152, TL5: ResNet50, TL6: VGG16, TL7: VGG19 using DC4.

**Figure 16 diagnostics-13-01954-f016:**
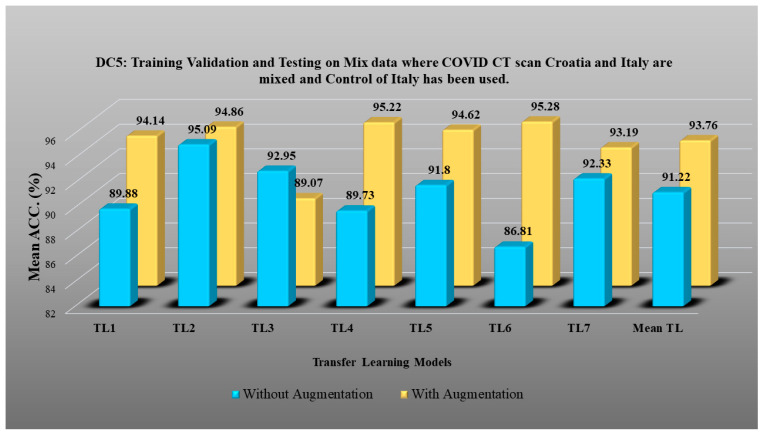
Comparison of mean TL accuracy with/without augmentation. TL1: EfficientV2M, TL2: InceptionV3, TL3: MobileNetV2, TL4: ResNet152, TL5: ResNet50, TL6: VGG16, TL7: VGG19 using DC5.

**Figure 17 diagnostics-13-01954-f017:**
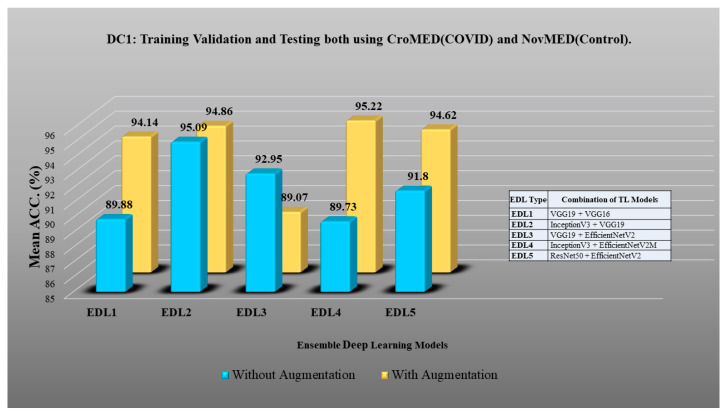
Comparison of mean EDL accuracy with/without augmentation. EDL1: VGG19 + VGG16, EDL2: InceptionV3 + VGG19, EDL3: VGG19 + EfficientNetV2M, EDL4: InceptionV3 + EfficientNetV2M, EDL5: ResNet50 + EfficientNetV2M using DC1.

**Figure 18 diagnostics-13-01954-f018:**
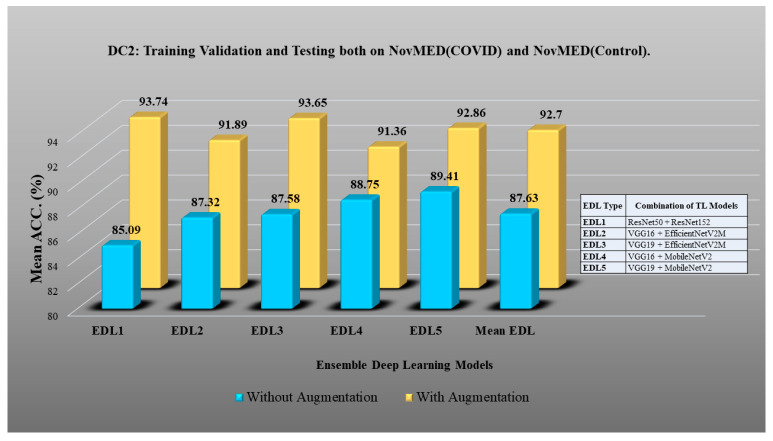
Comparison of mean EDL accuracy with/without augmentation. EDL1: ResNet50 + ResNet152, EDL2: VGG16 + EfficientNetV2M, EDL3: VGG19 + EfficientNetV2M, EDL4: VGG16 + MobileNetV2, EDL5: VGG19 + MobileNetV2 using DC2.

**Figure 19 diagnostics-13-01954-f019:**
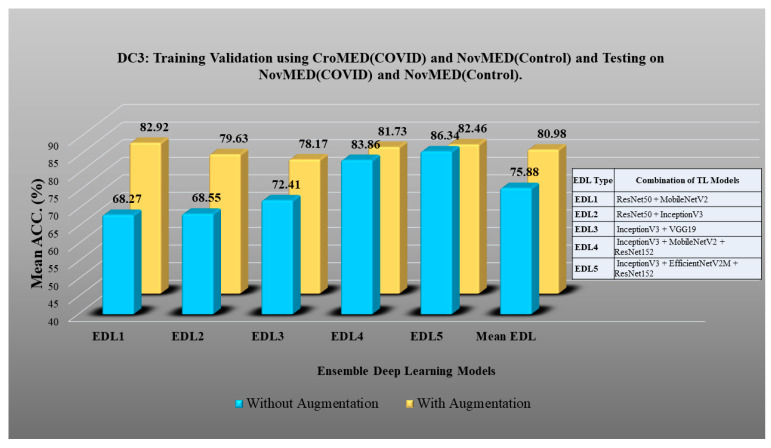
Comparison of mean EDL accuracy with/without augmentation. EDL1: ResNet50 + MobileNetV2, EDL2: ResNet50 + InceptionV3, EDL3: InceptionV3 + VGG19, EDL4: InceptionV3 + MobileNetV2 + ResNet152, EDL5: InceptionV3 + EfficientNetV2M + ResNet152 using DC3.

**Figure 20 diagnostics-13-01954-f020:**
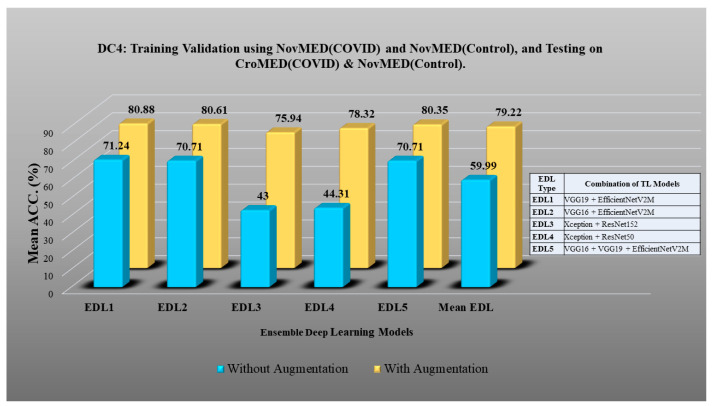
Comparison of mean EDL accuracy with/without augmentation. EDL1: VGG19 + EfficientNetV2M, EDL2: VGG16 + EfficientNetV2M, EDL3: Xception + ResNet152, EDL4: Xception + ResNet50, EDL5: VGG16 + VGG19 + EfficientNetV2M using DC4.

**Figure 21 diagnostics-13-01954-f021:**
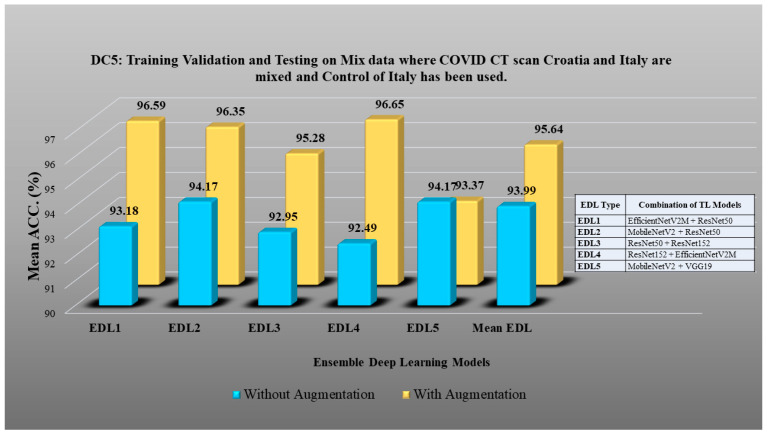
Comparison of mean EDL accuracy with/without augmentation. EDL1: EfficientNetV2M + ResNet50, EDL2: MobileNetV2 + ResNet50, EDL3: ResNet50 + ResNet 152, EDL4: ResNet152 + EfficientNetV2M, MobileNetV2 + VGG19 using DC5.

**Figure 22 diagnostics-13-01954-f022:**
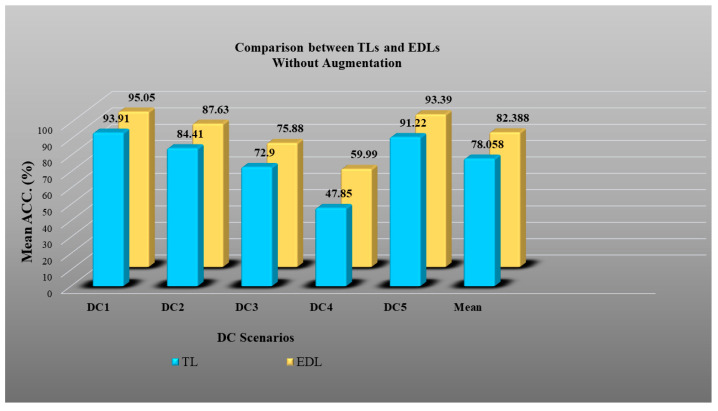
Accuracy plot of TL vs. EDL without augmentation for DC1, DC2, DC3, DC4, and DC5.

**Figure 23 diagnostics-13-01954-f023:**
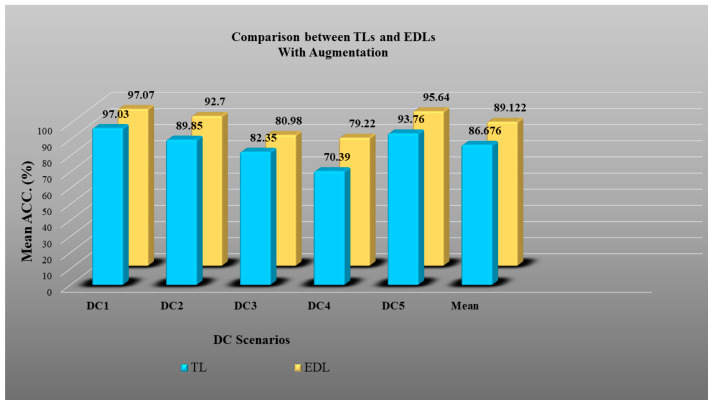
Accuracy plot of TL vs. EDL with augmentation for DC1, DC2, DC3, DC4, and DC5.

**Figure 24 diagnostics-13-01954-f024:**
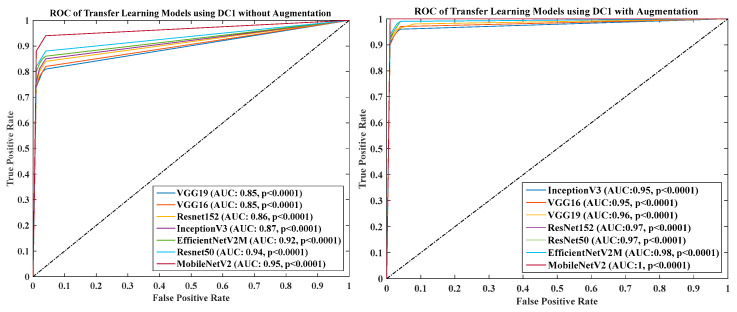
ROC of seven TLs using DC1 with augmentation and without augmentation.

**Figure 25 diagnostics-13-01954-f025:**
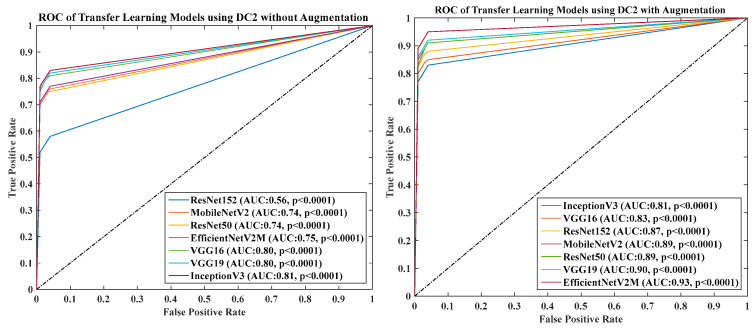
ROC of seven TLs using DC2 with augmentation and without augmentation.

**Figure 26 diagnostics-13-01954-f026:**
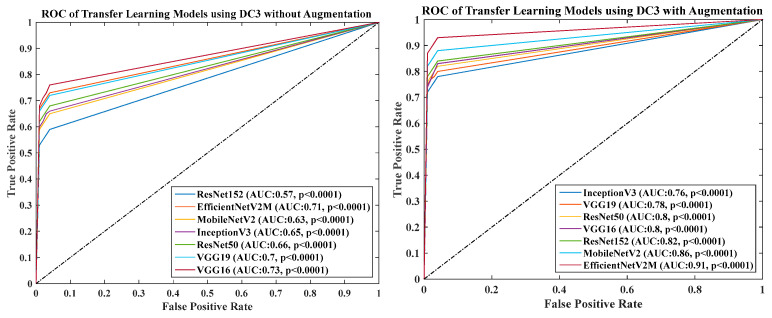
ROC of seven TLs using DC3 with augmentation and without augmentation.

**Figure 27 diagnostics-13-01954-f027:**
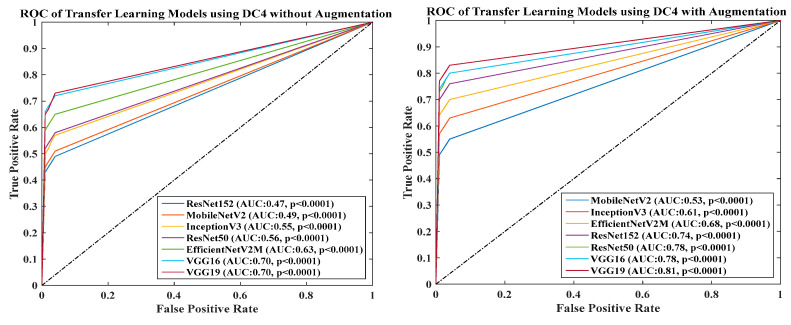
ROC of seven TLs using DC4 with augmentation and without augmentation.

**Figure 28 diagnostics-13-01954-f028:**
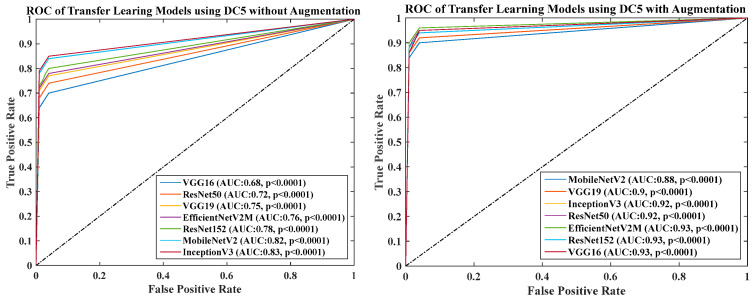
ROC of seven TLs using DC5 with augmentation and without augmentation.

**Figure 29 diagnostics-13-01954-f029:**
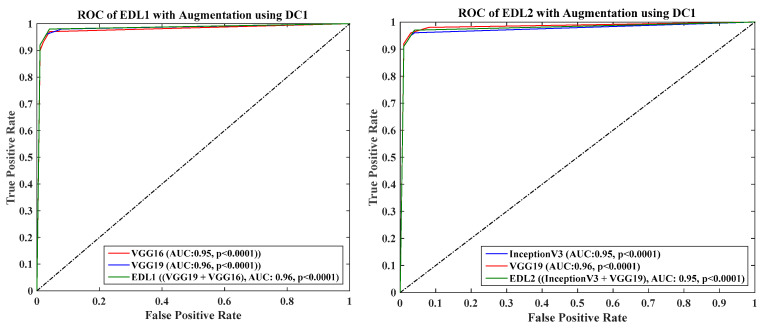
ROC of five EDLs using DC1 with augmentation.

**Table 1 diagnostics-13-01954-t001:** Summary of CroMED (COVID), NovMED (COVID), and NovMED (control) dataset information for the experiment.

SN	Dataset (Name)	Country of Origin	Patients	Images (Before Augmentation)	Images (After Augmentation)	Image Dimensions *	Training: Testing Split
1	CroMED (COVID)	Croatia	80	5396	10,792	512 × 512	K5
2	NovMED (COVID)	Novara, Italy	60	5797	11,594	768 × 768	K5
3	NovMED (Control)	Novara, Italy	12	1855	11,130	768 × 768	K5

* Pixel square; K5: 80% training and 20% testing data sets.

**Table 2 diagnostics-13-01954-t002:** Comparative TL statistical analysis using with/without augmentation on DC1.

TL Statistics on DC1											
	Without Augmentation						Balance + With Augmentation				
	TL Type	Mean ACC (%)	SD (%)	Mean PR	AUC (0–1)	*p*-Value	Mean ACC (%)	SD (%)	Mean PR	AUC (0–1)	*p*-Value
TL1	EfficientNetV2M	93.93	0.44	0.26	0.92	<0.0001	98.44	0.49	0.48	0.98	<0.0001
TL2	InceptionV3	93.65	0.39	0.19	0.87	<0.0001	93.97	0.49	0.47	0.95	<0.0001
TL3	MobileNetV2	97.93	0.42	0.23	0.95	<0.0001	99.99	0.49	0.5	1.00	<0.0001
TL4	ResNet152	92.9	0.38	0.18	0.86	<0.0001	96.98	0.49	0.47	0.97	<0.0001
TL5	ResNet50	97.1	0.41	0.22	0.94	<0.0001	97.44	0.49	0.48	0.97	<0.0001
TL6	VGG16	90.2	0.42	0.23	0.85	<0.0001	95.61	0.49	0.47	0.95	<0.0001
TL7	VGG19	91.72	0.39	0.19	0.85	<0.0001	96.8	0.49	0.48	0.96	<0.0001
		Mean ACC of all TLs: 93.91%					Mean ACC of all TLs: 97.03%				

**Table 3 diagnostics-13-01954-t003:** Comparative precision, recall and F1 score analysis of COVID and control classes with/without augmentation on DC1.

TL Statistics on DC1							
Without Augmentation							
	TL Type	COVID Precision	Control Precision	COVID Recall	Control Recall	COVID F1 Score	Control F1 Score
TL1	EfficientNetV2M	0.97	0.86	0.95	0.91	0.96	0.88
TL2	InceptionV3	0.93	1.00	1.00	0.78	0.96	0.88
TL3	MobileNetV2	0.97	1.00	1.00	0.92	0.99	0.96
TL4	ResNet152	0.91	1.00	1.00	0.72	0.95	0.84
TL5	ResNet50	0.94	0.98	0.99	0.81	0.97	0.89
TL6	VGG16	0.92	0.84	0.95	0.77	0.94	0.8
TL7	VGG19	0.97	0.81	0.93	0.92	0.95	0.86
**Balance + With Augmentation**							
	**TL Type**	**COVID Precision**	**Control Precision**	**COVID Recall**	**Control Recall**	**COVID F1 Score**	**Control F1 Score**
TL1	EfficientNetV2M	0.97	0.93	0.92	0.98	0.95	0.95
TL2	InceptionV3	0.99	1.00	1.00	0.99	1.00	1.00
TL3	MobileNetV2	1.00	1.00	1.00	1.00	1.00	1.00
TL4	ResNet152	0.99	1.00	1.00	0.99	0.99	0.99
TL5	ResNet50	0.99	1.00	1.00	0.99	1.00	1.00
TL6	VGG16	0.94	0.92	0.92	0.95	0.93	0.93
TL7	VGG19	0.92	0.95	0.95	0.92	0.93	0.93

**Table 4 diagnostics-13-01954-t004:** Comparative TL statistics analysis with/without augmentation on DC2.

TL Statistics on DC2											
	Without Augmentation						Balance + With Augmentation				
	TL Type	Mean ACC (%)	SD (%)	Mean PR	AUC (0–1)	*p*-Value	Mean ACC (%)	SD (%)	Mean PR	AUC (0–1)	*p*-Value
TL1	EfficientNetV2M	84.96	0.39	0.19	0.75	<0.0001	93.92	0.49	0.45	0.93	<0.0001
TL2	InceptionV3	90.84	0.35	0.15	0.81	<0.0001	90.04	0.49	0.44	0.81	<0.0001
TL3	MobileNetV2	85.22	0.37	0.16	0.74	<0.0001	89.77	0.49	0.49	0.89	<0.0001
TL4	ResNet152	78.16	0.2	0.04	0.56	<0.0001	87.4	0.49	0.52	0.87	<0.0001
TL5	ResNet50	79.47	0.44	0.27	0.74	<0.0001	93.53	0.49	0.46	0.89	<0.0001
TL6	VGG16	85.62	0.43	0.24	0.8	<0.0001	84.05	0.49	0.45	0.83	<0.0001
TL7	VGG19	86.66	0.42	0.22	0.8	<0.0001	90.3	0.49	0.43	0.9	<0.0001
		**Mean ACC of all TLs: 84.41%**					**Mean ACC of all TLs: 89.85%**				

**Table 5 diagnostics-13-01954-t005:** Comparative precision, recall, and F1 score analysis of COVID and control classes with/without augmentation in DC2-TL.

TL Statistics on DC2							
Without Augmentation							
	TL Type	COVID Precision	Control Precision	COVID Recall	Control Recall	COVID F1 Score	Control F1 Score
TL1	EfficientNetV2M	0.9	0.69	0.9	0.68	0.9	0.68
TL2	InceptionV3	0.87	0.38	0.64	0.71	0.74	0.5
TL3	MobileNetV2	0.87	0.8	0.96	0.54	0.81	0.65
TL4	ResNet152	0.88	0.48	0.76	0.68	0.82	0.56
TL5	ResNet50	0.89	0.53	0.81	0.69	0.85	0.6
TL6	VGG16	0.93	0.62	0.85	0.79	0.89	0.7
TL7	VGG19	0.9	0.6	0.85	0.7	0.87	0.65
**Balance + With Augmentation**							
	**TL Type**	**COVID Precision**	**Control Precision**	**COVID Recall**	**Control Recall**	**COVID F1 Score**	**Control F1 Score**
TL1	EfficientNetV2M	0.86	0.94	0.95	0.84	0.9	0.89
TL2	InceptionV3	0.85	0.83	0.83	0.84	0.84	0.84
TL3	MobileNetV2	0.73	1.00	1.00	0.61	0.84	0.76
TL4	ResNet152	0.8	0.78	0.78	0.8	0.79	0.79
TL5	ResNet50	0.83	0.83	0.84	0.83	0.84	0.83
TL6	VGG16	0.83	0.78	0.78	0.83	0.8	0.81
TL7	VGG19	0.84	0.88	0.89	0.83	0.86	0.85

**Table 6 diagnostics-13-01954-t006:** Comparative TL statistics analysis with/without augmentation on DC3.

TL Statistics on DC3											
	Without Augmentation						Balance + With Augmentation				
	TL Type	Mean ACC (%)	SD (%)	Mean PR	AUC (0–1)	*p*-Value	Mean ACC (%)	SD (%)	Mean PR	AUC (0–1)	*p*-Value
TL1	EfficientNetV2M	85.4	0.39	0.12	0.71	<0.0001	91.41	0.49	0.49	0.91	<0.0001
TL2	InceptionV3	67.6	0.48	0.38	0.65	<0.0001	76.43	0.48	0.63	0.76	<0.0001
TL3	MobileNetV2	67.3	0.47	0.35	0.63	<0.0001	86.75	0.48	0.38	0.86	<0.0001
TL4	ResNet152	78.1	0.18	0.03	0.57	<0.0001	82.28	0.49	0.59	0.82	<0.0001
TL5	ResNet50	67.2	0.49	0.4	0.66	<0.0001	80	0.48	0.61	0.8	<0.0001
TL6	VGG16	73.7	0.48	0.37	0.73	<0.0001	81	0.48	0.61	0.8	<0.0001
TL7	VGG19	71.2	0.48	0.38	0.7	<0.0001	78.63	0.49	0.56	0.78	<0.0001
		Mean ACC of all TLs: 72.90%					Mean ACC of all TLs: 82.35%				

**Table 7 diagnostics-13-01954-t007:** Comparative precision, recall, and F1 score analysis of COVID and control classes with/without augmentation in DC3.

TL Statistics on DC3							
	Without Augmentation						
	TL Type	COVID Precision	Control Precision	COVID Recall	Control Recall	COVID F1 Score	Control F1 Score
TL1	EfficientNetV2M	0.87	0.79	0.95	0.58	0.91	0.67
TL2	InceptionV3	0.84	0.41	0.69	0.63	0.76	0.5
TL3	MobileNetV2	0.82	0.4	0.72	0.55	0.77	0.46
TL4	ResNet152	0.77	1.00	1.00	0.14	0.87	0.25
TL5	ResNet50	0.85	0.41	0.68	0.65	0.75	0.5
TL6	VGG16	0.89	0.49	0.74	0.72	0.81	0.58
TL7	VGG19	0.87	0.46	0.72	0.7	0.79	0.55
**Balance + With Augmentation**							
	**TL Type**	**COVID Precision**	**Control Precision**	**COVID Recall**	**Control Recall**	**COVID F1 Score**	**Control F1 Score**
TL1	EfficientNetV2M	0.91	0.92	0.92	0.91	0.91	0.91
TL2	InceptionV3	0.86	0.71	0.63	0.9	0.72	0.79
TL3	MobileNetV2	0.8	0.98	0.99	0.75	0.88	0.85
TL4	ResNet152	0.89	0.78	0.74	0.91	0.8	0.84
TL5	ResNet50	0.88	0.75	0.69	0.91	0.78	0.82
TL6	VGG16	0.89	0.76	0.7	0.92	0.78	0.83
TL7	VGG19	0.82	0.76	0.72	0.85	0.77	0.8

**Table 8 diagnostics-13-01954-t008:** Comparative TL statistics analysis with/without augmentation on DC4.

TL Statistics on DC4											
	Without Augmentation						Balance + With Augmentation				
	TL Type	Mean ACC (%)	SD (%)	Mean PR	AUC (0–1)	*p*-Value	Mean ACC (%)	SD (%)	Mean PR	AUC (0–1)	*p*-Value
TL1	EfficientNetV2M	49.7	0.45	0.7	0.63	<0.0001	67.57	0.41	0.77	0.68	<0.0001
TL2	InceptionV3	45.6	0.47	0.66	0.55	<0.0001	60.52	0.33	0.87	0.61	<0.0001
TL3	MobileNetV2	28	0.26	0.92	0.49	<0.0001	52.68	0.21	0.95	0.53	<0.0001
TL4	ResNet152	35.2	0.42	0.75	0.47	<0.0001	73.92	0.46	0.68	0.74	<0.0001
TL5	ResNet50	39.7	0.4	0.78	0.56	<0.0001	78.41	0.48	0.6	0.78	<0.0001
TL6	VGG16	67.5	0.49	0.45	0.7	<0.0001	78.59	0.47	0.66	0.78	<0.0001
TL7	VGG19	69.4	0.49	0.41	0.7	<0.0001	81.05	0.48	0.63	0.81	<0.0001
		Mean ACC of all TLs: 47.85%					Mean ACC of all TLs: 70.39%				

**Table 9 diagnostics-13-01954-t009:** Comparative precision, recall, and F1 score analysis of COVID and control classes with/without augmentation in DC4.

TL Statistics on DC4							
Without Augmentation							
	TL Type	COVID Precision	Control Precision	COVID Recall	Control Recall	COVID F1 Score	Control F1 Score
TL1	EfficientNetV2M	0.93	0.31	0.37	0.91	0.52	0.47
TL2	InceptionV3	0.82	0.27	0.36	0.75	0.5	0.4
TL3	MobileNetV2	0.72	0.24	0.08	0.92	0.14	0.38
TL4	ResNet152	0.73	0.23	0.23	0.72	0.35	0.35
TL5	ResNet50	0.87	0.27	0.24	0.89	0.38	0.42
TL6	VGG16	0.9	0.41	0.64	0.77	0.75	0.53
TL7	VGG19	0.88	0.42	0.69	0.72	0.77	0.53
**Balance + With Augmentation**							
	**TL Type**	**COVID Precision**	**Control Precision**	**COVID Recall**	**Control Recall**	**COVID F1 Score**	**Control F1 Score**
TL1	EfficientNetV2M	0.92	0.61	0.4	0.97	0.56	0.74
TL2	InceptionV3	0.96	0.55	0.24	0.99	0.38	0.71
TL3	MobileNetV2	0.78	0.42	0.69	0.72	0.77	0.53
TL4	ResNet152	0.9	0.64	0.49	0.94	0.64	0.76
TL5	ResNet50	0.88	0.72	0.67	0.9	0.76	0.8
TL6	VGG16	0.94	0.71	0.62	0.96	0.75	0.81
TL7	VGG19	0.94	0.74	0.68	0.95	0.78	0.83

**Table 10 diagnostics-13-01954-t010:** Comparative TL statistics analysis with/without augmentation on DC5.

TL Statistics on DC5											
	Without Augmentation						Balance + With Augmentation				
	TL Type	Mean ACC (%)	SD (%)	Mean PR	AUC (0–1)	*p*-Value	Mean ACC (%)	SD (%)	Mean PR	AUC (0–1)	*p*-Value
TL1	EfficientNetV2M	89.9	0.32	0.12	0.76	<0.0001	94.14	0.47	0.33	0.93	<0.0001
TL2	InceptionV3	95.1	0.29	0.09	0.83	<0.0001	94.86	0.45	0.29	0.92	<0.0001
TL3	MobileNetV2	93	0.32	0.12	0.82	<0.0001	89.07	0.48	0.36	0.88	<0.0001
TL4	ResNet152	89.7	0.34	0.14	0.78	<0.0001	95.22	0.45	0.29	0.93	<0.0001
TL5	ResNet50	91.8	0.25	0.06	0.72	<0.0001	94.62	0.45	0.28	0.92	<0.0001
TL6	VGG16	86.8	0.31	0.11	0.68	<0.0001	95.28	0.45	0.28	0.93	<0.0001
TL7	VGG19	92.3	0.27	0.08	0.75	<0.0001	93.19	0.45	0.29	0.9	<0.0001
		Mean ACC of all TLs: 91.22%					Mean ACC of all TLs: 93.76%				

**Table 11 diagnostics-13-01954-t011:** Comparative precision, recall, and F1 score analysis of COVID and control classes with/without augmentation in DC5.

TL Statistics on DC5							
Without Augmentation							
	TL Type	COVID Precision	Control Precision	COVID Recall	Control Recall	COVID F1 Score	Control F1 Score
TL1	EfficientNetV2M	0.93	0.67	0.95	0.57	0.94	0.61
TL2	InceptionV3	0.95	0.98	1.00	0.66	0.97	0.79
TL3	MobileNetV2	0.95	0.79	0.97	0.69	0.96	0.73
TL4	ResNet152	0.94	0.64	0.94	0.63	0.94	0.64
TL5	ResNet50	0.92	0.93	0.99	0.45	0.95	0.61
TL6	VGG16	0.94	0.39	0.82	0.69	0.88	0.5
TL7	VGG19	0.93	0.88	0.99	0.53	0.96	0.66
**Balance + With Augmentation**							
	**TL Type**	**COVID Precision**	**Control Precision**	**COVID Recall**	**Control Recall**	**COVID F1 Score**	**Control F1 Score**
TL1	EfficientNetV2M	0.96	0.91	0.95	0.92	0.96	0.91
TL2	InceptionV3	0.94	0.98	0.99	0.86	0.96	0.92
TL3	MobileNetV2	0.94	0.81	0.89	0.88	0.92	0.84
TL4	ResNet152	0.94	0.99	0.99	0.88	0.97	0.93
TL5	ResNet50	0.93	0.99	1.00	0.85	0.96	0.91
TL6	VGG16	0.94	0.99	1.00	0.86	0.97	0.92
TL7	VGG19	0.92	0.95	0.98	0.84	0.95	0.89

**Table 12 diagnostics-13-01954-t012:** Comparative EDL statistics analysis with/without augmentation on DC1.

EDL Statistics on DC1										
EDL Type	Without Augmentation					Balance + With Augmentation				
	Mean ACC (%)	SD (%)	Mean PR	AUC (0–1)	*p*-Value	Mean ACC (%)	SD (%)	Mean PR	AUC (0–1)	*p*-Value
EDL1	92.82	0.4	0.2	0.87	<0.0001	96.89	0.49	0.47	0.96	<0.0001
EDL2	94.06	0.39	0.19	0.88	<0.0001	95.43	0.49	0.46	0.95	<0.0001
EDL3	94.48	0.41	0.21	0.9	<0.0001	98.26	0.49	0.48	0.98	<0.0001
EDL4	95.31	0.42	0.23	0.92	<0.0001	96.8	0.49	0.47	0.96	<0.0001
EDL5	98.62	0.42	0.24	0.97	<0.0001	97.99	0.49	0.48	0.98	<0.0001
	Mean ACC of all EDLs: 95.05%					Mean ACC of all EDLs: 97.07%				

**Table 13 diagnostics-13-01954-t013:** Comparative EDL statistics analysis with/without augmentation on DC2.

EDL Statistics on DC2										
	Without Augmentation					Balance + With Augmentation				
EDL Type	Mean ACC (%)	SD (%)	Mean PR	AUC (0–1)	*p*-Value	Mean ACC (%)	SD (%)	Mean PR	AUC (0–1)	*p*-Value
EDL1	85.09	0.36	0.15	0.73	<0.0001	93.74	0.49	0.46	0.93	<0.0001
EDL2	87.32	0.39	0.19	0.75	<0.0001	91.89	0.49	0.46	0.91	<0.0001
EDL3	87.58	0.39	0.19	0.8	<0.0001	93.65	0.49	0.45	0.93	<0.0001
EDL4	88.75	0.4	0.2	0.82	<0.0001	91.36	0.49	0.47	0.91	<0.0001
EDL5	89.41	0.39	0.19	0.82	<0.0001	92.86	0.49	0.46	0.92	<0.0001
	Mean ACC of all EDLs: 87.63%					Mean ACC of all EDLs: 92.70%				

**Table 14 diagnostics-13-01954-t014:** Comparative EDL statistics analysis with/without augmentation on DC3.

EDL Statistics on DC3										
	Without Augmentation					Balance + With Augmentation				
EDL Type	Mean ACC (%)	SD (%)	Mean PR	AUC (0–1)	*p*-Value	Mean ACC (%)	SD (%)	Mean PR	AUC (0–1)	*p*-Value
EDL1	68.27	0.48	0.38	0.66	<0.0001	82.92	0.49	0.52	0.8	<0.0001
EDL2	68.55	0.48	0.39	0.67	<0.0001	79.63	0.48	0.61	0.79	<0.0001
EDL3	72.41	0.48	0.37	0.71	<0.0001	78.17	0.49	0.58	0.78	<0.0001
EDL4	83.86	0.38	0.17	0.73	<0.0001	81.73	0.49	0.58	0.81	<0.0001
EDL5	86.34	0.36	0.15	0.75	<0.0001	82.46	0.49	0.59	0.82	<0.0001
	Mean ACC of all EDLs: 75.88%					Mean ACC of all EDLs: 80.98%				

**Table 15 diagnostics-13-01954-t015:** Comparative EDL statistics analysis using with/without augmentation on DC4.

EDL Statistics on DC4										
	Without Augmentation					Balance + With Augmentation				
EDL Type	Mean ACC (%)	SD (%)	Mean PR	AUC (0–1)	*p*-Value	Mean ACC (%)	SD (%)	Mean PR	AUC (0–1)	*p*-Value
EDL1	71.24	0.49	0.44	0.74	<0.0001	80.88	0.48	0.63	0.81	<0.0001
EDL2	70.71	0.49	0.47	0.76	<0.0001	80.61	0.47	0.65	0.8	<0.0001
EDL3	43	0.46	0.67	0.52	<0.0001	75.94	0.47	0.65	0.76	<0.0001
EDL4	44.31	0.45	0.7	0.56	<0.0001	78.32	0.48	0.61	0.78	<0.0001
EDL5	70.71	0.48	0.47	0.76	<0.0001	80.35	0.47	0.64	0.8	<0.0001
	Mean ACC of all EDLs: 59.99%					Mean ACC of all EDLs: 79.22%				

**Table 16 diagnostics-13-01954-t016:** Comparative EDL statistics analysis with/without augmentation on DC5.

EDL Statistics on DC5										
	Without Augmentation					Balance + With Augmentation				
EDL Type	Mean ACC (%)	SD (%)	Mean PR	AUC (0–1)	*p*-Value	Mean ACC (%)	SD (%)	Mean PR	AUC (0–1)	*p*-Value
EDL1	93.18	0.26	0.07	0.75	<0.0001	96.59	0.46	0.3	0.95	<0.0001
EDL2	94.17	0.29	0.09	0.81	<0.0001	96.35	0.46	0.3	0.94	<0.0001
EDL3	92.95	0.28	0.08	0.77	<0.0001	95.28	0.45	0.28	0.92	<0.0001
EDL4	92.49	0.3	0.1	0.78	<0.0001	96.65	0.46	0.3	0.95	<0.0001
EDL5	94.17	0.31	0.1	0.83	<0.0001	93.37	0.46	0.31	0.91	<0.0001
	Mean ACC of all EDLs: 93.39%					Mean ACC of all EDLs: 95.64%				

**Table 17 diagnostics-13-01954-t017:** Summary of paired *t*-test, Mann–Whitney, and Wilcoxon tests for TL models using five data combinations.

	TL Type	Paired *t*-Test	Mann–Whitney	Wilcoxon
TL1	EfficientNetV2M	*p* < 0.0001	*p* < 0.0001	*p* < 0.0001
TL2	InceptionV3	*p* < 0.0001	*p* < 0.0001	*p* < 0.0001
TL3	MobileNetV2	*p* < 0.0001	*p* < 0.0001	*p* < 0.0001
TL4	ResNet152	*p* < 0.0001	*p* < 0.0001	*p* < 0.0001
TL5	ResNet50	*p* < 0.0001	*p* < 0.0001	*p* < 0.0001
TL6	VGG16	*p* < 0.0001	*p* < 0.0001	*p* < 0.0001
TL7	VGG19	*p* < 0.0001	*p* < 0.0001	*p* < 0.0001

**Table 18 diagnostics-13-01954-t018:** Summary of paired *t*-test, Mann–Whitney, and Wilcoxon tests for EDL models using five data combinations.

EDL	Paired *t*-Test	Mann–Whitney	Wilcoxon
EDL1	*p* < 0.0001	*p* < 0.0001	*p* < 0.0001
EDL2	*p* < 0.0001	*p* < 0.0001	*p* < 0.0001
EDL3	*p* < 0.0001	*p* < 0.0001	*p* < 0.0001
EDL4	*p* < 0.0001	*p* < 0.0001	*p* < 0.0001
EDL5	*p* < 0.0001	*p* < 0.0001	*p* < 0.0001

**Table 19 diagnostics-13-01954-t019:** Transfer-learning-based models’ comparison.

SN	Yr	Author	Models	TL	Dataset	DS	Accu (%)	Pre (%)	Re (%)	F1 (%)	AUC (0–1)	*p*-Value	Cli. Val.	Sci Val.
1	2021	Alshazly et al. [28]	DenseNet201	TL	COVID-CT	746	92.9	91.3	-	92.5	0.93	-	✗	✗
2	2021	Alshazly et al. [28]	DenseNet169	TL	COVID-CT	746	91.2	88.1	-	90.8	0.91	-	✗	✗
3	2021	Cruz et al. [29]	DenseNet161	TL	COVID-CT	746	82.76	85.39	77.55	81.28	0.89	-	✗	✗
4	2021	Cruz et al. [29]	VGG16	TL	COVID-CT	746	81.77	79.05	84.69	81.77	0.9	-	✗	✗
5	2022	Shaik et al. [30]	MobileNetV2	TL	SARS-CoV-2	2482	97.38	97.41	97.35	97.38	0.97	-	✗	✗
6	2022	Shaik et al. [30]	MobileNetV2	TL	COVID-CT	746	88.67	88.5	88.61	88.55	0.88	-	✗	✗
7	2022	Huang et al. [31]	EfficientNetV2M	TL	COVID-CT	7463	95.66	95.67	95.58	95.65	0.97	-	✗	✗
8	2023	Xu et al. [32]	DenseNet121	TL	COVIDx-CT 2A	3745	99.44	99.89	-	-	-	-	✗	✗
9	2023	Proposed	MobileNetV2 (Best)	TL	Dataset1	5797 1855	99.99	100	100	100	1.00	<0.0001	✗	✓

**Table 20 diagnostics-13-01954-t020:** Ensemble-deep-learning-based models’ comparison.

SN	Yr	Name	Models	EDL	Dataset	DS	Accu (%)	Pre (%)	Re (%)	F1 (%)	AUC (0–1)	*p*-Value	Cli. Val.	Sci. Val.
1	2021	Pathan et al. [33]	ResNet50, AlexNet, VGG19, DenseNet, Inception V3	EDL	COVID-CT	746	97.00	97.00	97.00		0.97	-	✗	✗
2	2021	Kundu et al. [34]	VGG-11, GoogLeNet, SqueezeNet v1, Wide ResNet-50-2	EDL	SARS-CoV-2	2482	98.93	98.93	98.93	98.93	0.98	-	✗	✗
3	2021	Tao et al. [35]	AlexNet, GoogleNet, ResNet	EDL	COVID-CT	2933	99.05	-	-	98.59		0	✗	✗
4	2021	Cruz et al. [29]	VGG16, ResNet50, Wide-ResNet50, DenseNet161/169, InceptionV3	EDL	COVID-CT	746	90.7	93.27	89.69	94.05	0.95	-	✗	✗
5	2022	Shaik et al. [30]	VGG16, ResNet50,	EDL	COVID-CT	746	91.33	91.29	91.16	91.22	0.91	-	✗	✗
6	2022	Khanibadi et al. [138]	Naïve Bays, Support Vector Machine	EDL	COVID-CT	746	93.00	92.7	93.5	94.4	0.94	-	✗	✗
7	2022	Lu et al. [139]	Self-Supervised model with Loss function	EDL	COVID-CT	746	94.3	0.94	0.93	0.94	0.98	<0.0001	✗	✗
8	2022	Huang et al. [31]	EfficientNet B0-B4 and V2M	EDL	COVID-CT	7463	98.84	98.87	98.93	98.92	0.99	0	✗	✗
9		Proposed	ResNet152, MobileNetV2	EDL	Dataset1 *	7652	99.99	100	100	100	100	<0.0001	✗	✓

* CroMED (COVID), NovMED (Control).

## Data Availability

Not applicable.

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
