# Peer review of "Ensemble Deep Learning Derived from Transfer Learning for Classification of COVID-19 Patients on Hybrid Deep-Learning-Based Lung Segmentation: A Data Augmentation and Balancing Framework"

_diagnostics, 2023, doi:10.3390/diagnostics13111954_

Round 1

Reviewer 1 Report

This is a detailed study comprising three datasets, seven transfer learning-based models, and five deep ensemble learning models performing the classification of subjects with and without COVID-19. There were two hypotheses for this study: (i) EDL is better than individual TL and (ii) augmentation of data provides better stratification compared to using data without augmentation. The authors have provided sufficient analysis to prove their hypotheses. The authors have also provided different experiments to check the effect of EDL on TL. A comparative analysis with previous literature also indicates the potential of this study for COVID-19 classification. There are a few minor points on which the authors' attention should be needed. These are as follows:

1.       Database information is ambiguous: It is not clear how many datasets were used for this study design. The authors talk about CroMED dataset, NovMED dataset, COVID (Croatia) dataset, COVID (ITA) dataset, COVID (Italy) dataset, and NovMED (Control) dataset. It would be more appropriate to provide details about different datasets, the number of patients, and the number of images in a tabular format.

2.       Augmentation and Class Distribution: It is not clear how many images were used to train the models. Also, what was the total number of images before and after augmentation? Imbalance data handling: Authors should highlight the class distribution before and after augmentation.

Addressing these points, the manuscript may be accepted for publication in the journal.

Sentence ambiguity: The manuscript needs to be checked for grammar and English. Some of the sentences are ambiguous and can be redrafted. E.g., “…we have designed False to Top layers of all models…

Author Response

Dear Reviewer ,

thank you once again for the inputs and helping us in improving the quality of the 
manuscript. Authors of the manuscript.

Reviewer 2 Report

The authors have done extensive results comparison which is quite interesting and contributing to the research community. 

However, the graphs explanation can be enhanced further. 

In the methodology section, the explanation on the Segmentation techniques needs to be elaborated. 

Author Response

(The authors gave the same response as above.)

Reviewer 3 Report

This paper presents an automated and accurate COVID-19 diagnosis system that verifies that ensemble deep learning (EDL) is superior to deep transfer learning (TL) in both non-augmented and augmented frameworks. Seven TL models and five EDL models were adopted for classification on ResNet- UNet-based segmentation of CT scans on five different kinds of data combinations (DC). The paper discusses a topic that was extensively investigated the past three years. Lots of papers have been published regarding automated COVID-19 diagnosis where several studies included EDL models. The related work lacks plenty of related works discussing automated COVID-19 diagnosis.

Abstract

Several previous studies constructed frameworks for automatic COVID-19 diagnosis based on EDL, TL, and data augmentation. They also proved that EDL is better than DL. Moreover, plenty of studies verified that TL improves the classification/detection accuracy. What is the novelty in your proposed framework? And what is the difference between your model and previous models?

Introduction

Why did the author perform binary classification only? Why did the authors not differentiate between different types of pneumonia and COVID-19?

Related work:

Several important papers that employed EDL, TL and data augmentation are missing from this section. I advise the authors to add these related studies.

1.     RADIC: A tool for diagnosing COVID-19 from chest CT and X-ray scans using deep learning and quad-radiomics. Chemometrics and Intelligent Laboratory Systems (2023): 104750.

2.      Coronavirus covid-19 detection by means of explainable deep learning." Scientific Reports 13.1 (2023): 462

3.     Diagnosis of COVID-19 using CT scan images and deep learning techniques. Emergency Radiology 2021;28:497–505.

4.     A wavelet-based deep learning pipeline for efficient COVID-19 diagnosis via CT slices, Applied Soft Computing

5.     Ensemble deep learning and internet of things-based automated COVID-19 diagnosis framework.Contrast Media & Molecular Imaging 2022 (2022).

6.    A computer-aided diagnostic framework for coronavirus diagnosis using texture-based radiomics images. Digital Health

7.     Classification of COVID-19 chest CT images based on ensemble deep learning." Journal of Healthcare Engineering 2021 (2021).

8.     "The ensemble deep learning model for novel COVID-19 on CT images." Applied soft computing 98 (2021): 106885.

9.      FUSI-CAD: Coronavirus (COVID-19) diagnosis based on the fusion of CNNs and handcrafted features.

10. MULTI-DEEP: A novel CAD system for coronavirus (COVID-19) diagnosis from CT images using multiple convolution neural networks.

11. "EDNC: ensemble deep neural network for Covid-19 recognition." Tomography 8, no. 2 (2022): 869-890.

12. Deep learning-based CAD system for COVID-19 diagnosis via spectral-temporal images." Proceedings of the 12th International Conference on Information Communication and Management. 2022.

Also, the authors should discuss their limitations which motivated them to propose a new approach. Please discuss the advantages and limitations of these techniques and add their results.

Methodology

What were the inclusion and exclusion criteria for the dataset acquisition?

Please highlight the novelty of the proposed method as there are many papers that were previously proposed for COVID-19 diagnosis using EDL, TL, and data augmentation.

The resolution of figure 4 is low. Please improve.

What are the hyperparameters of the DL and EDL models?

How were these hyperparameters selected or fine-tuned?

Author Response

(The authors gave the same response as above.)

Reviewer 4 Report

The article discusses the use of ensemble deep learning (EDL) and deep transfer learning (TL) in the classification of COVID-19 disease versus control using lung computed tomography (CT) scans. The authors hypothesize that EDL is superior to TL in both non-augmented and augmented frameworks due to challenges in the variability of CT scans. The system used a cascade of quality control, hybrid deep learning, TL-based classification followed by EDL, and was implemented on five different kinds of data combinations taken from Croatia and Italy. The results show that EDL outperformed TL in both unbalanced and un-augmented as well as balanced and augmented datasets in both seen and unseen paradigms. All statistical tests proved positive for reliability and stability. The study suggests that EDL can be an effective solution for accurate COVID-19 disease classification using lung CT scans.

The abstract should provide more details on the specific methods and techniques used in the study, such as the specific deep learning models and the criteria for selecting the data combinations from Croatia and Italy. The article should provide a more detailed background and motivation for the study, explaining why the accuracy of COVID-19 disease classification using lung CT scans is important and how EDL and TL can address the challenges. The article should provide more information on the data used in the study, such as the characteristics of the patients and the CT scans, to enable better interpretation of the results. The statistical tests used to assess the reliability and stability of the results should be explained more clearly and in greater detail. The article should provide a more in-depth discussion of the limitations and future directions of the research, such as potential sources of bias and how the findings could be extended to larger and more diverse datasets.

Author Response

(The authors gave the same response as above.)

Reviewer 5 Report

Authors have proposed a deep learning-based system for lung segmentation. I have some comments below:

1. The manuscript is not organized well. Please revise the format.

2. Please provide more explanation on the legends of the figures. For example, in Figure 1, what are these subfigures?

3. What are the core contributions? Authors have used a lot of techniques from literatures, but which of them are original contributions?

4. The equations have some mathematical typos, please revise it.

5. The sample size in the paper is really small. Can the result of statistical testing really prove the reliability?

There are some typos and grammar errors in the manuscript. The writing of the manuscript needs to be revised. 

Author Response

(The authors gave the same response as above.)

Round 2

Reviewer 3 Report

The authors have addressed most of the comments properly.

Please rewrite the abstract to highlight the motivation, novelty and contributions.

Author Response

Dear Reviewer, thank you once again for the inputs and helping us in improving the quality of the manuscript. Authors of the manuscript. 

Old Abstract

     Abstract: Background and Motivation: Lung computed tomography (CT) techniques has been adopted in the intensive care unit (ICU) for COVID-19 disease versus control classification due to its high-resolution imaging. The diagnosis lacks an accurate design of Artificial Intelligence (AI)-based solution due to challenges in glass ground opacity, consolidations, and crazy paving lesions in CT scans, leading to variabilities. We hypothesize that ensemble deep learning (EDL) is superior to deep transfer learning (TL) in both non-augmented and augmented frameworks. Methodology: The system consists of a cascade of quality control, hybrid deep learning, TL-based classification followed by EDL. Seven TL models and five EDL models were adopted for classification on ResNet-UNet-based segmentation of CT scans on five different kinds of data combinations (DC) taken from Croatia (80 patients) and Italy (72 patients COVID and 30 patients Controls), totaling 12,000 CT slices. These paradigms were implemented in non-augmented and augmented framework. As part of generalization, the system was tested on Unseen data, and statistically tested for reliability and stability. Results: Using K5 (80:20) protocol, the balanced and augmented dataset, all five DC datasets yielded an improved TL mean accuracy by 3.32%, 6.56%, 12.96%, 47.1%, and 2.78%, respectively. The five EDL systems showed an improvement in accuracy by 2.12%, 5.78%, 6.72%, 32.05%, and 2.40%, thus validating our hypothesis. All statistical tests proved positive for reliability and stability. Conclusion: EDL showed superior performance to TL systems for both (a) unbalanced and un-augmented and (b) balanced and augmented dataset in (i) seen and (ii) unseen paradigms, demonstrating both hypotheses.

New Abstract

 Abstract: Background and Motivation: Lung computed tomography (CT) techniques are high-resolution and are well adopted in the intensive care unit (ICU) for COVID-19 disease versus control classification. Most artificial intelligence (AI) systems do not undergo generalization and are typically overfitted.  Such trained AI systems are not practical for clinical settings, and therefore, do not give accurate results when executed on unseen data sets. We hypothesize that ensemble deep learning (EDL) is superior to deep transfer learning (TL) in both non-augmented and augmented frameworks. Methodology:  The system consists of a cascade of quality control, ResNet-UNet-based hybrid deep learning for lung segmentation, and seven models using TL-based classification followed by five types of EDL’s.  To prove our hypothesis, five different kinds of data combinations (DC) were designed using a combination of two multicenter cohorts: Croatia (80 COVID), and Italy (72 COVID and 30 Controls), leading to 12,000 CT slices. As part of generalization, the system was tested on unseen data and statistically tested for reliability/stability. Results: Using K5 (80:20) cross-validation protocol, the balanced and augmented dataset, all five DC datasets yielded an improved TL mean accuracy by 3.32%, 6.56%, 12.96%, 47.1%, and 2.78%, respectively. The five EDL systems showed an improvement in accuracy by 2.12%, 5.78%, 6.72%, 32.05%, and 2.40%, thus validating our hypothesis. All statistical tests proved positive for reliability and stability. Conclusion: EDL showed superior performance to TL systems for both (a) unbalanced and un-augmented and (b) balanced and augmented datasets for both (i) seen and (ii) unseen paradigms, validating both our hypotheses.

Reviewer 5 Report

The revision has improved the quality of the manuscript. 

Author Response

Thank you very much for improving the readability of the manuscript.